

# Six years of continuous carbon isotope composition measurements of methane in Heidelberg (Germany) — a study of source contributions and comparison to emission inventories

Antje Hoheisel[1] and Martina Schmidt[1]

[1]Institute of Environmental Physics, Heidelberg University, Heidelberg, Germany

**Correspondence:** Antje Hoheisel (antje.hoheisel@iup.uni-heidelberg.de)

**Abstract.** $\delta(^{13}CH_4)$ and the mole fraction of $CH_4$ have been measured continuously since April 2014 using a cavity ring-down spectroscopy (CRDS) analyser in Heidelberg, Germany. This 6-year time series shows an increasing trend of $(6.8 \pm 0.3)$ $nmol\,mol^{-1}\,a^{-1}$ for the $CH_4$ mole fraction between 2014 and 2020. $\delta(^{13}CH_4)$ decreases by $(-0.028 \pm 0.002)\text{‰}\,a^{-1}$ over this time period.

In this study, seasonal variations and trends of $CH_4$ emissions in the catchment area of Heidelberg are analysed using three approaches by applying the Miller-Tans method to atmospheric measurements on different time scales. The mean $\delta^{13}C$ isotopic source signature for the Heidelberg catchment area is $(-52.5 \pm 0.3)\text{‰}$ (moving Miller-Tans approach). In all three approaches, there is no significant trend in the monthly mean source signature over the last six years. However, more depleted source signature values occur in summer. This annual cycle in $^{13}C$-$CH_4$ sources, with a peak-to-peak amplitude of $-6.2\text{‰}$, can only be partially explained by seasonal variations in $CH_4$ emissions from heating. Additional seasonal variations probably occur in biogenic $CH_4$ emissions from waste water, landfills or dairy cows.

Furthermore, the source contributions derived from atmospheric measurements are used to evaluate the $CH_4$ emissions reported by two emission inventories: the Emissions Database for Global Atmospheric Research (EDGAR v6.0) and the inventory of the State Institute for the Environment Baden-Württemberg (LUBW - Landesanstalt für Umwelt Baden-Württemberg). The mean $\delta(^{13}CH_4)$ source signature determined from the LUBW inventory agrees well with the result from atmospheric measurements. However, the signature determined from EDGAR v6.0 data is less depleted by about $7\text{‰}$. Thus, EDGAR v6.0 seems to overestimate $CH_4$ emissions from more enriched sources.

## 1   Introduction

One of the most challenging problems of our time is global warming. To limit the negative impacts associated with climate change, the 2015 UN Paris Agreement on Climate Change has set the goal to limit the mean global temperature increase to below $2\,°C$, preferably to $1.5\,°C$, compared to pre-industrial level (UNFCCC, 2015). In 2021 the United States, the European Union, and other countries launched the Global Methane Pledge with the goal to reduce global methane emissions. This initiative recognised the short lifetime of methane ($CH_4$) of only 9.1 to 11.8 years (IPCC, 2021), allowing for a more rapid effect on atmospheric $CH_4$ mole fraction after reducing $CH_4$ emissions.



On a global scale several studies have analysed atmospheric carbon isotope ratios in methane, in addition to $CH_4$ mole fractions to constrain emission budgets and to explain observed atmospheric trends in mole fraction (e.g. Nisbet et.al, 2016, 2019; Schaefer, 2016, 2019 and Lan et al., 2021). This is possible, since each source type has a different isotopic signature depending on the production processes and origin.

$CH_4$ is emitted from anthropogenic and natural sources, which are grouped in three different categories according to the pro-
duction processes. Most depleted $CH_4$ ($-55‰$ to $-70‰$) is specified as biogenic and is produced under anaerobic conditions due to degradation of organic matter. Typical biogenic $CH_4$ sources are wetlands, ruminants, landfills and wastewater treatment plants. Thermogenic $CH_4$, like that in natural gas, is formed on geological time scales out of organic matter and is more enriched with values between $-25‰$ and $-45‰$. Most enriched $CH_4$ ($-13‰$ to $-25‰$) is formed during the incomplete combustion of organic matter, such as biomass burning, and is characterised as pyrogenic. (IPCC, 2013)

The knowledge of the spatial and temporal variation of $CH_4$ emissions around the world, and their composition from different types of sources, is important to reduce $CH_4$ emissions effectively and to understand the influence of different $CH_4$ sources on climate change. Also on a local and regional scale, the measurement of atmospheric $\delta(^{13}C, CH_4)$, hereafter abbreviated as $\delta(^{13}CH_4)$, provides information about the contribution of different emission sectors to the total $CH_4$ emissions. Traditionally, $\delta(^{13}CH_4)$ in the atmosphere is measured by collecting weekly flask or sample bags and analysing them with isotope ratio
mass spectrometry (Miller et al., 2002; Fischer et al., 2006; Zazzeri et al., 2015; Röckmann et al., 2016). This method was used by Levin et al. (1999), who analysed and evaluated bi-weekly atmospheric samples in Heidelberg in the 1990s. With new measurement techniques such as continuous flow isotope ratio mass spectrometry (IRMS), quantum cascade laser absorption spectroscopy (QCLAS) or cavity ring-down spectroscopy (CRDS), the $\delta(^{13}CH_4)$ values in ambient air can be measured continuously and with high temporal resolutions from a few seconds up to minutes (Eyer et al., 2016; Röckmann et al., 2016;
Hoheisel et al., 2019).

There is a growing number of studies analysing atmospheric measurements of $\delta(^{13}CH_4)$ and of $CH_4$ mole fractions with high temporal resolution. Assan et al. (2018) analysed $\delta(^{13}CH_4)$ measurements near industrial sites and Röckmann et al. (2016), as well as Menoud et al. (2020), studied $\delta(^{13}CH_4)$ in rural areas in the Netherlands. $CH_4$ measured at urban stations, however, originates from heterogeneously distributed sources including waste management, natural gas distribution systems,
heating, transport and agriculture. The corresponding emissions vary strongly in their isotopic $^{13}C$-$CH_4$ composition, and make the analysis and interpretation of $CH_4$ emissions in cities more difficult (Menoud et al., 2021). However, isotope studies with high-resolution measurements can also contribute to revealing possible inconsistencies in emission inventories in urban areas. By analysing a 2-year time series of $\delta(^{13}CH_4)$ in London, Saboya et al. (2022) demonstrated that emissions from natural gas leaks are underestimated in both the UK National Atmospheric Emissions Inventory (UK NAEI) and the Emissions Database
for Global Atmospheric Research (EDGAR).

At the urban station Heidelberg, the atmospheric $CH_4$ mole fraction and isotopic composition $\delta(^{13}CH_4)$ have been measured continuously with a CRDS analyser since 2014. This measurement device enables the analysis of $CH_4$ and $\delta(^{13}CH_4)$ at high temporal resolution of a few seconds. To our knowledge, our time series is the longest in situ $\delta(^{13}CH_4)$ record, with high temporal resolution, reported to date. $CH_4$ emissions around Heidelberg originate from different sources due to the urban region





with rural surroundings. The regional emission inventory from the State Institute for the Environment Baden-Württemberg (LUBW - Landesanstalt für Umwelt Baden-Württemberg) classified the $CH_4$ emissions for 2016 for the Heidelberg region to the following main sectors: agriculture (30 %), waste management (30 %) and natural gas distribution systems (28 %) (LUBW, 2016).

In this study, a continuous six-year time series of the atmospheric $CH_4$ mole fraction and $\delta(^{13}CH_4)$ at the urban station
Heidelberg is analysed to identify and understand seasonal and long-term variabilities of regional and local $CH_4$ sources. Different approaches, such as the moving Miller-Tans approach, are used to determine the contribution of different sectors to $CH_4$ emissions in the catchment area of Heidelberg. These results are then compared to a regional emission inventory provided by LUBW, and the emission database EDGAR v6.0. Thus, atmospheric measurements are used to verify the estimated contribution of the different emission sectors to $CH_4$ emissions in the emission inventories.

## 2 Methods


### 2.1 Site description

Heidelberg ($\approx 159\,000$ inhabitants) is located in the south-west of Germany and in the north of the state Baden-Württemberg. It is situated in the Upper Rhine Plain on the edge of the low mountain range Odenwald (Fig. 1). Therefore, the north-east is less urban and more forested. More agricultural and urban areas are in the Upper Rhine Plain from the north-west to south-
east. The industrial cities of Mannheim ($\approx 312\,000$ inhabitants) and Ludwigshafen ($\approx 172\,000$ inhabitants) are 15 km to 20 km north-west of Heidelberg. Due to its location within industrial, urban, agricultural and rural areas, $CH_4$ emissions measured in Heidelberg can originate from biogenic (e.g. dairy cows, waste water treatment plants), thermogenic (e.g. natural gas), and even pyrogenic (e.g. traffic) sources.

### 2.2 Experimental setup

Since April 2014, a cavity ring-down spectroscopy (CRDS) G2201-i analyser (Picarro, Inc., Santa Clara, CA) has been con- tinuously measuring the dry air mole fraction of $CH_4$ and its $^{13}C/^{12}C$ ratio in ambient air with a temporal resolution of a few seconds. The intake for these ambient air measurements is located on the roof of the Institute for Environmental Physics (IUP - Institut für Umweltphysik) in Heidelberg, 30 m above ground. Several studies have shown that the internal water correction, especially for $\delta(^{13}CH_4)$, is insufficient for this type of analyser (Rella et al., 2015; Hoheisel et al., 2019) and air drying is
required for precise measurements. Thus, a cold trap cooled by a cryostat dries the air before it enters the CRDS analyser through a 16-way rotary valve (model: EMT2CSD16UWE, Valco Vici, Switzerland). The gas flow through the analyser is typ- ically about 80 ml min$^{-1}$ and is monitored by an electronic flow meter (model: 5067-0223, Agilent Technologies, Inc., Santa Clara, CA). Every five hours, the ambient air measurement is interrupted to analyse calibration and quality control gases for 20 minutes each. The schematic of the laboratory setup is shown in Fig. 2.



### 2.3   Data treatment

The G2201-i analyser records $CH_4$ and the isotopic composition $\delta(^{13}CH_4)$ every $3.7\,s$. These high temporal resolution data are averaged to one-minute values. Before analysing these minutely $CH_4$ and $\delta(^{13}CH_4)$ values of ambient air, artefacts, outliers and invalid data are identified and flagged. These include periods of technical problems, work on the experimental setup such as replacing the cold trap, and the first five minutes after a change of sample gas to account for flushing of the cavity.

The one-minute $CH_4$ mole fractions and the isotopic composition of $CH_4$ are calibrated with a single-point calibration using the calibration measurements carried out every five hours. In August 2019, the calibration cylinder had to be replaced (see Table A1). The $CH_4$ mole fraction measurements are reported on the WMO X2004A scale (Dlugokencky et al., 2005) in $nmol\,mol^{-1} = 10^{-9}$ (nanomole per mole of dry air). The measurements of the isotopic compositions of $CH_4$ are traced to the Vienna Pee Dee Belemnite (VPDB) isotopic scale (Sperlich et al., 2016). Hence, in 2014 and 2019, the calibration cylinders

were analysed with the gas chromatography (GC) system in Heidelberg (Levin et al., 1999) and the $\delta(^{13}CH_4)$ values were measured by the Stable Isotope Laboratory at Max Planck Institute for Biogeochemistry (MPI-BGC) in Jena.

### 2.4   Instrumental performance

The instrumental precision of the analyser was determined in 2013 and 2019 by performing measurements on different gas cylinders for at least $12\,h$ each. The Allan standard deviation determined from these measurements can be used as a measure

of the repeatability of a measurement over a certain period of time. The Allan standard deviation of atmospheric $CH_4$ is below $0.11\,nmol\,mol^{-1}$ even for the high-resolution one-minute data. For an averaging interval of $15\,min$, corresponding to the calibration and target gas measurements, and $CH_4$ mole fractions between $1922\,nmol\,mol^{-1}$ and $2004\,nmol\,mol^{-1}$, the Allan standard deviation of $CH_4$ and $\delta(^{13}CH_4)$ is $0.08\,nmol\,mol^{-1}$ and $0.24\%o$, respectively (see Fig. A1). The long-term reproducibility of the CRDS G2201-i analyser, i.e. the standard deviation of the target gas measurements performed between

2014 and 2020, is $0.2\,nmol\,mol^{-1}$ for $CH_4$ and $0.3\%o$ for $\delta(^{13}CH_4)$ (see Fig. A2).

Six intercomparison cylinders with air samples from Neumayer Station in Antarctica were measured with our CRDS G2201 analyser to validate the measurement accuracy. These cylinders had already been analysed by the MPI-BGC within the framework of an interlaboratory comparison (Umezawa et al., 2018). The average difference in $\delta(^{13}CH_4)$ between our results and the MPI-BGC measurements is $(0.02 \pm 0.05)\%o$ (see Table A2).

## 3   Results and Discussion

### 3.1   Continuous $CH_4$ mole fraction and $\delta(^{13}CH_4)$ measurements

Atmospheric $CH_4$ mole fraction and $\delta(^{13}CH_4)$ were measured continuously with a CRDS analyser in Heidelberg between April 2014 and May 2020. Figure 3 shows the daily mean $CH_4$ mole fractions, which vary between $1890\,nmol\,mol^{-1}$ and $2310\,nmol\,mol^{-1}$, with higher values in winter than in summer. The corresponding isotopic composition $\delta(^{13}CH_4)$ ranges

from $-49\%o$ to $-47\%o$.



The digital filter curve fitting programme CCGCRV[1] developed by Kirk Thoning (Earth System Group, Earth System Laboratory (CCG/ESRL), NOAA, Thoning et al. 1989) is applied to the monthly average data to analyse the trend and annual cycle of $CH_4$ and $\delta(^{13}CH_4)$. Between 2014 and 2020, the $CH_4$ mole fraction increases by $(6.8 \pm 0.3)\,\text{nmol}\,\text{mol}^{-1}\,\text{a}^{-1}$ and $\delta(^{13}CH_4)$ shows a decreasing trend of $(-0.028 \pm 0.002)\text{‰}\,\text{a}^{-1}$. Furthermore, $CH_4$ and $\delta(^{13}CH_4)$ show strong mean annual cycles (right panel of Fig. 3). The maximum of the mean $CH_4$ mole fraction occurs in late autumn (November). In winter and spring, the mole fraction decreases slightly until it reaches a minimum in late summer (June to July). The amplitude (peak-to-peak height) is $78\,\text{nmol}\,\text{mol}^{-1}$ in $CH_4$. The annual cycle in atmospheric $\delta(^{13}CH_4)$ has a mean amplitude of $0.4\text{‰}$. Less enriched $\delta(^{13}CH_4)$ values of $-48.3\text{‰}$ occur in early autumn (September to October), with the most enriched values of $-47.9\text{‰}$ in spring (April to May).

In addition to the trend and the annual cycle, the $CH_4$ mole fraction and $\delta(^{13}CH_4)$ show diurnal variations. The mean diurnal cycles for different seasons are presented in Fig. 4. In the afternoon (15-16 UTC), the overnight increase in the $CH_4$ mole fraction begins due to the lower mixing height. After sunrise, the mole fraction decreases strongly due to radiation-induced mixing and thus an increase of the mixing height. The mean diurnal cycles show strong seasonal differences with larger variations in summer ($52\,\text{nmol}\,\text{mol}^{-1}$) and weaker ones in winter ($21\,\text{nmol}\,\text{mol}^{-1}$). Since the diurnal cycle is strongly driven by the sun, the earlier sunrise and later sunset in summer compared to winter is additionally noticeable by the earlier decrease of $CH_4$ in the morning and the later increase in the afternoon. The diurnal variations of $\delta(^{13}CH_4)$ show slightly larger amplitudes in summer ($0.18\text{‰}$) and autumn ($0.16\text{‰}$) than in winter ($0.09\text{‰}$) and spring ($0.12\text{‰}$). The lowest $\delta(^{13}CH_4)$ values occur around 7 to 10 UTC. $\delta(^{13}CH_4)$ increases during the day to maximum values between 18 and 21 UTC, before decreasing at night. It seems that in summer, the depletion in $\delta(^{13}CH_4)$ in the morning is slightly stronger than in the other seasons.

## 3.2 Comparison of $\delta(^{13}CH_4)$ with background and former measurements

In Heidelberg, the $CH_4$ mole fraction and $\delta(^{13}CH_4)$ were measured with a GC- IRMS system and from bi-weekly integral flask samples between 1992 and 1997 (Levin et al., 1999). Since the previous $CH_4$ mole fractions were reported on the CMDL83 scale, we take into account that the $CH_4$ mole fractions measured on the new WMO 2004 scale are a factor of $(1.0124 \pm 0.0007)$ larger (Dlugokencky et al., 2005). Figure 5 shows $CH_4$ and $\delta(^{13}CH_4)$ from the two time periods (1992-1998, 2014-2020) for which $\delta(^{13}CH_4)$ measurements were done in Heidelberg. In addition to the Heidelberg measurements, data from the background station Mace Head Observatory (Lan et al., 2022; Michel et al., 2022) are shown. The isotopic composition measured at Mace Head by the Institute of Arctic and Alpine Research (INSTAAR) of the University of Colorado has to be subtracted by an offset of $0.28\text{‰}$ to take into account the inter-comparison offset among the laboratories INSTAAR and MPI-BGC (Umezawa et al., 2018).

Again the curve fitting program CCGCRV is applied to the monthly mean values to determine trends and seasonal variabilities. The observed increasing trend in Heidelberg between April 2014 and June 2020 is only slightly smaller than the one in Mace Head. This is different in the 1990s, where the $CH_4$ mole fraction did not follow the increasing trend observed at the background station Izaña (Levin et al., 1999) or Mace Head. Furthermore, the continental $CH_4$ excess at Heidelberg (Heidel-

---

[1]CCGCRV: https://www.esrl.noaa.gov/gmd/ccgg/mbl/crvfit/index.html and ftp://ftp.cmdl.noaa.gov/user/thoning/ccgcrv/



berg data minus Mace Head data) strongly decreased between the 1990s and recent years (2014-2020) to $(70 \pm 3)\,\mathrm{nmol\,mol^{-1}}$,

which is only half of the value from the 1990s. These observations can be explained by a change in the emission rate in the catchment area of Heidelberg. In the studies by Levin et al. (2011, 2021) the $CH_4$ fluxes in Heidelberg are calculated with the Radon-Tracer method. They found a $30\,\%$ reduction of $CH_4$ emissions between 1996 and 2004 and no further systematic trend thereafter. In the 1990s, the $\delta(^{13}CH_4)$ values in Heidelberg decreased strongly with $-0.14\,‰\,\mathrm{a^{-1}}$, while samples from Izaña only show trends which are more than a factor of three smaller (Levin et al., 1999). This difference in the $\delta(^{13}CH_4)$ trends

points to a change in the composition of $CH_4$ emissions in the catchment area of Heidelberg. Levin et al. (1999) attribute this change to a reduction of $CH_4$ emissions from fossil sources (mainly coal mining) and from cattle breeding. The situation is different for recent measurements (2014 to 2020). The current Heidelberg data only show a small trend in $\delta(^{13}CH_4)$ which is similar to the one observed at Mace Head. Therefore, the $CH_4$ source mixture in Heidelberg seems to be relatively constant during the last years.

### 3.3 Isotopic carbon signature of $CH_4$ sources calculated with atmospheric measurements

$CH_4$ sources contributing to the atmospheric $CH_4$ mole fraction have different $\delta^{13}C$ isotopic source signatures depending on their origin and production process. These isotopic source signatures can range from $-13\,‰$ to $-70\,‰$ (Sherwood et al., 2021; Menoud et al., 2022). Therefore, the measured atmospheric $\delta(^{13}CH_4)$ value strongly depends on the $CH_4$ source mixture from regional and local sources. That makes it possible to analyse the $CH_4$ sources in the Heidelberg catchment based on the

measured atmospheric $CH_4$ mole fraction in combination with the observed atmospheric isotopic composition $\delta(^{13}CH_4)$. In most cases, an increase in atmospheric $CH_4$ mole fraction will be caused by a mixture of $CH_4$ emitted from different sources. Thus, from the atmospheric measurements, one usually does not obtain information about a single source, but the average isotopic signature of several contributing sources depending on their respective emission rate.

### 3.3.1 Determination of mean $\delta^{13}C$ isotopic source signatures

In this study we use the Miller-Tans method (Miller and Tans, 2003) in combination with the York fit (York et al., 2004) to determine the mean $\delta^{13}C$ isotopic source signature in the catchment area of Heidelberg. This method is applied to the one-minute averages of $CH_4$ and $\delta(^{13}CH_4)$ for which the Allan standard deviation is used as a measure of uncertainty. The Miller-Tans method uses the linear relationship between $\delta_{\mathrm{obs}} \cdot C_{\mathrm{obs}}$ and $C_{\mathrm{obs}}$, where $C$ and $\delta$ refer to $CH_4$ and $\delta(^{13}CH_4)$:

$$\delta_{\mathrm{obs}} \cdot C_{\mathrm{obs}} = C_{\mathrm{bg}} \cdot (\delta_{\mathrm{bg}} - \delta_{\mathrm{s}}) + \delta_{\mathrm{s}} \cdot C_{\mathrm{obs}}. \tag{1}$$

Here, the indices obs, bg and s denote observed, background and source values. The York fit was chosen as this method minimises the weighted distance between the data points and the fitted line, taking into account uncertainties in both x and y-coordinates. Tests have shown that for our application there is no difference between the Miller-Tans or the Keeling plot (Keeling, 1958, 1961) methods when using the York fit. The compatibility of these methods was also shown by Zobitz et al. (2006) for $CO_2$ and Hoheisel et al. (2019) for $CH_4$. The uncertainty of the source signature determined with the Miller-Tans

method and the York fit strongly depends on the precision of the analyser and the peak height of $CH_4$ above background





(Hoheisel et al., 2019). To achieve accurate results for the mean $\delta^{13}$C isotopic source signatures, we apply two criteria to our data: the $CH_4$ range has to be larger than $100\,\mathrm{nmol\,mol^{-1}}$ and the fit error on the slope of the regression line has to be smaller than $2.5\,‰$.

Different approaches are tested for the choice of time scale (month, night, event) for which the mean $\delta^{13}$C isotopic source signature for Heidelberg should be calculated. Depending on the time scale, the Miller-Tans method is applied to different data subsets (each month, each night, moving interval). Larger time intervals of one month have the advantage that the $CH_4$ mole fractions cover a large range, which increases the precision of the results of the regression line. On the other hand, uncertainties occur since the background is probably not constant over the entire time period, which can be assumed for shorter time intervals of a few hours. The three most promising approaches used in this study are the monthly, the night-time and the moving Miller-Tans approach. In the monthly approach, the Miller-Tans method is applied to the one-minute average data of each month of each year. In the night-time approach, the Miller-Tans method is applied to the one-minute average data between 17 and 7 CET. This approach uses the night-time increase in the $CH_4$ mole fraction caused by the accumulation of $CH_4$ emissions in the lower boundary layer. Therefore, we determine the mean $\delta^{13}$C isotopic source signature of the contributing $CH_4$ sources for each night. In order to achieve meaningful results, only nocturnal data sets that fulfil our two criteria ($CH_4$ range $>100\,\mathrm{nmol\,mol^{-1}}$, regression fit error for the slope $<2.5\,‰$) are used. This is the case for $21\,\%$ of the night data sets. We can therefore determine the mean $\delta^{13}$C isotopic source signature in the catchment area of Heidelberg for 460 nights.

Due to the high temporal resolution of our $CH_4$ mole fraction and $\delta(^{13}CH_4)$ measurements, we can go one step further and determine the $\delta^{13}$C isotopic source signatures of events with a moving Miller-Tans approach similar to the moving Keeling plot or moving Miller-Tans methods used by Röckmann et al. (2016), Menoud et al. (2020), Assan et al. (2018) or Saboya et al. (2022). Since we are interested in short-term events, a time window with a fixed length of one hour is shifted over the one-minute average data set with time steps of one minute. Thus, for each minute $t_i$, the mean $\delta^{13}$C isotopic source signature is calculated from a one-hour time period centred on $t_i$ using the Miller-Tans method and the York fit. Again only those results which fulfil our two criteria of a $CH_4$ range larger than $100\,\mathrm{nmol\,mol^{-1}}$ during the time window and a fit error of the slope smaller than $2.5\,‰$ are used. If these criteria for $t_i$ are not achieved, the result for $t_i$ calculated with a time window one hour longer is used. This continues until both criteria are fulfilled or the length of the time window reaches 12 hours. If the criteria are still not met for the 12 hours time interval, the result is excluded. With the moving Miller-Tans approach, we achieve results for $18\,\%$ of the one-minute average data. To take into account that several of the mean isotopic source signatures determined for each minute may describe the same event, an average is taken over all values directly adjacent in time. Thus, the mean $\delta^{13}$C isotopic source signatures of 769 events are determined for the six years between April 2011 and May 2020.

### 3.3.2 Monthly averages and annual cycle of the mean $\delta^{13}$C isotopic source signatures

Figure 6a shows the monthly averaged values of the mean $\delta^{13}$C isotopic signatures of the $CH_4$ sources in the Heidelberg catchment area, which were determined using the monthly (black), night-time (blue) and moving Miller-Tans (red) approaches. The monthly mean $\delta^{13}$C isotopic source signatures vary between $-61.5\,‰$ and $-42.3\,‰$ and show similar results for the three different approaches. The average mean $\delta^{13}$C isotopic source signature of $CH_4$ in Heidelberg for the whole time period of six



years is $(-52.5 \pm 0.3)‰$ (mean $\pm$ standard error of the mean), calculated with the moving Miller-Tans approach. The result from the night-time approach is $(-52.3 \pm 0.4)‰$ and does not differ significantly from the moving Miller-Tans approach. The result from the monthly approach is $(-53.9 \pm 0.3)‰$ and is only slightly less enriched than the results from the other two approaches. Thus, the average mean isotopic source signature of $CH_4$ is more depleted than the mean $\delta(^{13}CH_4)$ value in the atmosphere in Heidelberg $(-48.07 \pm 0.02)‰$. This indicates a strong influence from biogenic $CH_4$ sources, such as

waste management and agriculture, in the catchment area of Heidelberg. In comparison, the mean isotopic source signatures determined for two five month measurement campaigns in more rural areas in the Netherlands, where ruminants are a main $CH_4$ source, were $(-60.8 \pm 0.2)‰$ (Röckmann et al., 2016) and $(-59.55 \pm 0.13)‰$ (Menoud et al., 2020). Looking at other studies in urban areas, Menoud et al. (2021) reported an overall source signature of $-48.7‰$ in Krakow (Poland, 6-month campaign), and Saboya et al. (2022) calculated a median isotopic source signature of $-41.6‰$ for London (UK, 2.7 years),

indicating that the primary $CH_4$ sources in London are natural gas leaks. The mean $\delta^{13}C$ isotope signature in Heidelberg thus shows a contribution from enriched sources such as natural gas, heating, and even traffic from the Heidelberg urban area in addition to biogenic emissions. However, neither of these sources appear to be the only main emitter. This is consistent with the emission inventory of the State Institute for the Environment Baden-Württemberg (LUBW, 2016) for the Heidelbeg area, which reports one third of the emissions each from natural gas leakage, the waste sector, or agriculture (see Fig. 8 in section 3.4.1).

Between 2014 and 2020, no significant trend is detectable in the monthly mean $\delta^{13}C$ isotopic source signatures obtained from all three approaches. Therefore, we assume that the general composition of $CH_4$ emissions in the Heidelberg catchment area has not changed or has changed only slightly during this period. This finding is different to a former study by Levin et al. (1999) from the 1990s. They found a change in the $\delta(^{13}CH_4)$ source signature from $(-47.4 \pm 1.2)‰$ in 1992/1993 to $(-52.9 \pm 0.4)‰$ in 1995/1996 and attribute this change to a reduction of $CH_4$ emissions from fossil sources (mainly coal

mining) and from cattle breeding.

Moreover, a commonality between the mean $\delta^{13}C$ isotopic source signatures calculated with the different approaches is that a strong annual cycle with more depleted values in the summer months can be noticed (Fig. 6b). The annual cycles calculated with all three approaches show most depleted source signatures in June. From June to October the $\delta^{13}C$ isotopic source signatures increase to more enriched values and stay relatively constant until April. Between April and June a strong

decrease to more depleted values is visible. This annual cycle clearly indicates that in summer the $CH_4$ emissions have a larger biogenic share compared to the rest of the year. When analysing each year individually, the majority have a detectable annual cycle, and it is therefore a very well-defined signal that does not arise from one or two very pronounced annual cycles.

### 3.3.3   Mean $\delta^{13}C$ isotopic source signatures of individual nights and events

An advantage of the night-time and moving Miller-Tans approach compared to the monthly approach is that the mean $\delta^{13}C$

isotopic source signature of individual nights or events can be studied. Figure 7a shows the histogram of the mean $\delta^{13}C$ isotopic source signatures of 460 individual nights calculated with the night-time approach, and Figure 7b displays a similar histogram using the mean $\delta^{13}C$ isotopic source signatures for the 769 events determined by the moving Miller-Tans approach. Most of the $CH_4$ emissions during one night or event are a mixture from several sources and cannot be attributed to one particular source.





When separating the night-time and event source signatures into winter/spring (Nov to Apr) and summer/autumn (May to Oct), a shift in the mean $\delta^{13}$C isotopic source signature of approximately $2.5\text{\textperthousand}$ is noticeable. More depleted mean $\delta^{13}$C isotopic source signatures of $(-53.5 \pm 0.4)\text{\textperthousand}$ or $(-53.6 \pm 0.3)\text{\textperthousand}$ occur in summer/autumn and more enriched ones of $(-51.1 \pm 0.5)\text{\textperthousand}$ or $(-51.0 \pm 0.4)\text{\textperthousand}$ in winter/spring for the night-time or the moving Miller Tans approach, respectively (Fig. 7). This annual cycle is also described in section 3.3.2. Both approaches additionally have in common that our criteria are fulfilled for fewer nights or events in winter than in summer. Only $41\%$ to $43\%$ of the determined $\delta^{13}$C isotopic source signatures occur between Nov and Apr. Since the diurnal variations are usually lower in winter than in summer, more night-time increases or events have ranges below the chosen threshold of $100\,\text{nmol}\,\text{mol}^{-1}$ and are therefore excluded.

Furthermore, we determined the diurnal cycle for the mean $\delta^{13}$C isotopic source signatures calculated with the moving Miller-Tans approach. However, the year to year variations are too strong compared to the possible mean diurnal cycle to get reliable results and to exclude the possibility that the noticeable diurnal variations are only an artefact of the averaging. Even though we can analyse the $\delta^{13}$C isotopic source signature at time scales below individual months, the precision of our analyser is still too low to interpret diurnal variations. Although, the development of new instruments with better precision of isotope measurements will soon make this possible.

### 3.3.4 Discussion of different approaches

The average mean $\delta^{13}$C isotopic source signatures of $CH_4$ and the annual cycles in Heidelberg calculated with the moving Miller-Tans approach or the night-time approach from the whole six-year time period show no significant differences. This indicates that the composition of $CH_4$ sources in Heidelberg is the same during day and night or that the emissions during the night-time increase contribute most in the moving Miller-Tans approach, too.

The monthly approach results in similar monthly mean $\delta^{13}$C isotopic source signatures and a similar annual cycle to the other two approaches. The average mean source signature is, however, approximately $1.4\text{\textperthousand}$ less enriched than the results from the moving Miller-Tans and the night-time approaches (Fig. 6 a). The reason for this difference cannot be conclusively resolved in this study. One possibility is that this difference can be caused by the assumption of a constant background over the entire month or the fact that all one-minute average data points of the month contribute to the determined source signature. In the night-time and the Miller-Tans approaches nights and time periods which do not fulfil our criteria are discarded, which can exclude small pollution events. As all data points are used in the monthly approach, the small events also contribute to the mean $\delta^{13}$C isotopic source signature. Another explanation can be, that the considered $CH_4$ emissions in the monthly, night-time and moving Miller-Tans approaches represent different catchment areas. $CH_4$ emissions from more distant sources show lower and more temporally extended $CH_4$ peaks in the measured time series than emissions from local and regional sources. In the analysis of small time intervals of several hours, more distant emissions can be excluded by the selection criteria. Thus, the night-time and moving Miller-Tans approach probably consider more distant emissions less often than local and regional ones. Furthermore, at night, the footprint of Heidelberg is smaller than during the day. In 2018, around $47\%$ of the surface influence calculated with the Stochastic Time-Inverted Lagrangian Transport (STILT) model (Lin et al., 2003 and Kountouris et al., 2018) for the station Heidelberg is within $50\,\text{km}$ at night (time of the day: 18 to 3), but within $100\,\text{km}$ during the day



(time of the day: 6 to 15). For these calculations the STILT footprint tools[2] and the STILT jupyter notebook service[3] were used. Thus, the monthly approach, which includes daytime data, represents a larger catchment area than the night-time approach.

Different $CH_4$ sources have different isotopic source signatures, which depend on the production process of $CH_4$. The isotopic source signatures of several sources in the surroundings of Heidelberg are characterised in Hoheisel et al. (2019). Biogenic $CH_4$ emitted from livestock, landfills, and wastewater treatment is more depleted. Thermogenic $CH_4$ from the gas distribution system is less depleted (see Table 1). Other studies such as Levin et al. (1999), Menoud et al. (2021), and Zazzeri et al. (2017) report isotopic source signatures from combustion processes for traffic, industry, and energy for buildings ( see

Table 1). This pyrogenic $CH_4$ is even less depleted than thermogenic $CH_4$. Since the measurement site is located in an urban area, the nearby $CH_4$ sources are more often natural gas leaks, wastewater, traffic, or emissions from energy for buildings. These $CH_4$ emissions are on average less depleted. The more distant sources tend to be in rural areas, so that emissions from landfills and livestock are more prominent. These biogenic emissions are more depleted. This agrees well with the more depleted mean $\delta^{13}C$ isotopic source signature of $CH_4$ calculated with the monthly approach, in comparison to the night-time approach.

We tested the robustness of the monthly, night-time and moving Miller-Tans approaches by varying the selection criteria. The $CH_4$ range was set to be $100 \, \text{nmol} \, \text{mol}^{-1}$, $150 \, \text{nmol} \, \text{mol}^{-1}$ or $200 \, \text{nmol} \, \text{mol}^{-1}$, and the threshold for the fit error of the slope was changed from 2.5‰ over 5‰ to 10‰. All determined monthly mean source signatures show similar results, with an annual cycle containing more biogenic values in summer. The monthly values vary on average between 0.1‰ and 0.8‰, with standard deviations between 1‰ and 3‰. Therefore, we choose the $CH_4$ range of $100 \, \text{nmol} \, \text{mol}^{-1}$ as threshold to include

more data sets and 2.5‰ as threshold for the fit error of the slope, and thus the uncertainty of the source signature, to still assure precise results.

    Furthermore, several automatic approaches to identify the nocturnal increases for each night in the time series were tested. The determined monthly averaged $\delta^{13}C$ isotopic source signatures did not vary strongly between the automatic approaches and the one using the fixed time window. Since the automatic approaches did not correctly identify the $CH_4$ increase for all nights,

we chose the same fixed time interval between 17 and 7 CET for the nightly increase of $CH_4$ for each day. Also, varying the fixed time interval does not lead to any relevant changes in the monthly averaged mean $\delta^{13}C$ isotopic source signatures. In addition, we tested a more common method for the moving Miller-Tans approach starting with a 12 hours time window. Then the time interval is reduced in hourly steps when our two criteria are not fulfilled. There is no significant difference between the monthly averaged mean $\delta^{13}C$ isotopic source signatures of the two moving Miller-Tans scenarios.

To conclude, all three approaches have their advantages depending on the temporal and spatial range we are interested in. We have shown that the monthly approach is a good and easy solution to determine the monthly mean source signature and deviates only slightly from the more specific night-time and moving Miller-Tans approach. Especially for remote stations which only observe small diurnal variations in $CH_4$ this method is a good option, when night-time and moving Miller-Tans approaches struggle with the low variations. We tested the monthly approach at the mountain station Schauinsland (1205 m a.s.l.) operated

by the German Environment Agency (UBA - Umweltbundesamt) to determine the mean $\delta^{13}C$ isotopic signature of $CH_4$ for

---

[2]STILT footprint tools: https://www.icos-cp.eu/data-services/tools/stilt-footprint

[3]STILT jupyter notebook service: https://www.icos-cp.eu/data-services/tools/jupyter-notebook





two measurement campaigns of one month. In the summer campaign the mean source signature is $(-60.3 \pm 0.7)$‰ and in the winter campaign $(-56.9 \pm 0.4)$‰. The larger influence of biogenic emissions in summer can also be seen at the Schauinsland station.

## 3.4 Comparison of CH$_4$ source contribution with different emission inventories

Emission inventories are based on bottom-up methods which involve statistical data about emitters, such as animal population or the amount and type of combusted fuel, and specific emission factors that quantify the emissions from different source categories (IPCC, 2006). Both, statistical data and emission factors, can have large uncertainties, for instance, due to unknown and unaccounted sources or high spatial and temporal variability. In addition to national emission inventories, regional emission inventories for each county are reported on a yearly basis, for example by the State Institute for the Environment Baden-

Württemberg (LUBW, 2016). Other emission inventories, such as the Emissions Database for Global Atmospheric Research (EDGARv6.0, Crippa et al., 2021), go one step further and aim to provide accurate annual emissions for different source types covering the entire globe. The different emission inventories can show, though, strong deviations in the amount and composition of emissions for the same area. Therefore, it is important to verify the reported greenhouse gas emissions given by emission inventories on a global, a national as well as a regional scale. Only then can the intended reduction of greenhouse

gases be confirmed and, if necessary, the mitigation strategy adapted.

In this study, the measurements of the atmospheric CH$_4$ mole fraction and the isotopic composition $\delta(^{13}CH_4)$ were used to calculate a mean $\delta^{13}C$ isotopic source signature and its annual cycle for the catchment area of Heidelberg (Sect. 3.3). In the following section, these results are compared to two different emission inventories to constrain their estimated emissions and to explain the noticed annual cycle in the mean $\delta^{13}C$ isotopic source signature determined for the catchment area of Heidelberg.

### 3.4.1 Emission inventories

The first emission inventory used in this study is provided by the State Institute for the Environment Baden-Württemberg (LUBW, 2016) and the second is the Emissions Database for Global Atmospheric Research (EDGAR v6.0, Crippa et al., 2021). Since the measurements in Heidelberg were carried out at low elevation about 30 m above ground and within the city, the atmospheric CH$_4$ mole fraction measurements are most strongly influenced by local and regional sources. The LUBW

provides detailed information about CH$_4$ emissions depending on different CH$_4$ categories for the cities of Heidelberg (HD) and Mannheim (MA), the county Rhein-Neckar-Kreis (RNK), and the state Baden-Württemberg (BW) for the reference year 2016 (see Fig. 8).

EDGAR v6.0[4] estimates CH$_4$ emissions from different categories for $0.1° \times 0.1°$ grid cells covering the whole world. In addition to annual sector-specific gridmaps, monthly sector-specific gridmaps are also provided for the years 2000 to 2018.

Emissions for the Heidelberg, Mannheim, and Rhein-Neckar-Kreis areas are determined from the monthly sector-specific gridmaps using all grid cells which are at least partly within the borders of the respective county. Thereby, the emissions from each cell are weighted in the summation according to the percentage of overlap between the cell and the county and are then

---

[4]EDGAR v6.0: https://edgar.jrc.ec.europa.eu/index.php/dataset_ghg60





added up for each year. The $CH_4$ emissions provided by EDGAR v6.0 for the years 2014 to 2018 vary between $12731\,\mathrm{t\,a^{-1}}$ and $13685\,\mathrm{t\,a^{-1}}$ in the Heidelberg area (including HD, MA and RNK), and seem to decrease slightly by $7\,\%$. The average emission
for the whole time period is $(13319 \pm 163)\,\mathrm{t\,a^{-1}}$.

Figure 8 shows the emissions for the Heidelberg area per section for LUBW (2016) and EDGAR (2014-2018). The sectors which contribute most are natural gas, waste treatment and livestock farming. For the Heidelberg area (HD,MA,RNK) the average emissions determined by EDGAR v6.0 are 3.4 times larger than $CH_4$ emissions provided by LUBW ($3915\,\mathrm{t\,a^{-1}}$). Both inventories report comparable $CH_4$ emissions from livestock farming (1.1 times larger emissions by EDGAR v6.0 than LUBW),
but strong differences occur for emissions from the waste treatment and waste incineration sector (3.5 times), the natural gas sector (4.9 times) and the energy for buildings sector (4.5 times). EDGAR v6.0 reports $CH_4$ emissions from waste incineration, which are comparable to the emissions from waste water treatment plants. These emissions are not reported separately by the LUBW.

The city of Mannheim forms a connected urban area with the city of Ludwigshafen and is only separated by the river
Rhine. Several industrial companies such as BASF are located there, especially near the river. In the EDGAR v6.0 inventory, strong $CH_4$ emissions occur in the two grid cells on the border between Mannheim and Ludwigshafen for the industry, gas, oil and waste treatment sectors. Thus, $CH_4$ emissions determined from the EDGAR v6.0 inventory for Mannheim can include emissions from Ludwigshafen. In these grid cells, $CH_4$ emissions from waste treatment or the power industry sector can be assigned primarily to sites in Mannheim. However, the emissions from combustion for the manufacturing sector as well as the
natural gas and oil sector cannot be separated so easily and could therefore lead to larger differences to the LUBW inventory. Unfortunately, to our knowledge, there is no sector-separated $CH_4$ inventory for Ludwigshafen that could be included in the LUBW inventory. However, the distribution of emissions at the border of the areas cannot explain the whole deviation. Indeed, the $CH_4$ emissions for all of Baden-Württemberg are still 1.5 times larger in EDGAR v6.0 than reported by LUBW. Again strong differences occur for the waste treatment and waste incineration sector (4.0 times larger emissions by EDGAR v6.0 than
LUBW) as well as the energy for buildings sector (3.9 times).

The differences between the reported $CH_4$ emissions by EDGAR v6.0 and LUBW are probably partly caused by differences in the statistical data, especially by different assumptions for the emission factors used to estimate the $CH_4$ emissions from different sectors. This is supported by the fact that the amount of emissions from sectors, such as livestock farming, with well studied emission factors and accurate statistical data are comparable for both inventories. $CH_4$ emissions estimated by
EDGAR v5.0 for Germany have an uncertainty of only $16\,\%$ for the agriculture sector, while the uncertainty for the waste sector is $43\,\%$ (Solazzo et al., 2021). These values are estimated for the $CH_4$ emissions of Germany. The uncertainty of individual or several grid cells can be even larger. The LUBW does not report uncertainties of the $CH_4$ emissions.

## 3.5  Mean $\delta^{13}$C isotopic signature of $CH_4$ sources in the Heidelberg area

The two emission inventories of LUBW and EDGAR v6.0 report $CH_4$ emissions depending on source sectors. By attributing
a source specific $\delta^{13}$C isotopic signature to the emissions of each sector, the mean $\delta^{13}$C isotopic signature of $CH_4$ sources in the Heidelberg area can be determined. The isotopic signatures for each source sector are chosen, if possible, from results of



measurement campaigns in the catchment area of Heidelberg (Hoheisel et al., 2019, Levin et al., 1993). Table 1 summarises the $\delta^{13}$C isotopic source signatures used for the different sectors. Despite intensive literature research we have not been able to find any publications describing $\delta^{13}$C for $CH_4$ emitted by waste incineration in the way we needed them to calculate the

mean $\delta^{13}$C isotopic source signature. Thus, we adopted the $^{13}$C composition of waste incineration reported by Widory et al. (2006). This is possible, since no strong isotopic fractionation is noticeable during the combustion for $CO_2$ and we assume that no strong fractionation of $^{13}$C occurs for $CH_4$, either.

The mean $\delta^{13}$C isotopic source signature for the Heidelberg area determined using the LUBW (2016) inventory is $-52‰$. The result calculated from the average EDGAR v6.0 data for the years 2014 to 2018 for the Heidelberg area is $-46‰$. The

uncertainty of the determined source signatures is $2‰$ and it is calculated from the variations in the $\delta^{13}$C isotopic signatures of the emission sectors. Since no uncertainties are reported for the $CH_4$ emissions in the LUBW inventory or the grid cells in the EDGAR v6.0 inventory, their impact on the determined mean source signature could not be taken into account.

A large difference of $6‰$ between the mean source signature determined from LUBW and EDGAR v6.0 data occurs and is caused by the differences in the relative source mixture. On the right side in Fig. 8, the relative amount of $CH_4$ emissions per

sector is shown for the Heidelberg area. Biogenic $CH_4$, which is most depleted, contributes most in the LUBW inventory from livestock farming and waste treatment giving $30\%$ each. In the EDGAR v6.0 inventory, only $10\%$ and $22\%$ of anthropogenic $CH_4$ is emitted by livestock farming and waste treatment in the Heidelberg area. At the same time, much more thermogenic and even pyrogenic $CH_4$, which is more enriched, is emitted in the EDGAR v6.0 (2014-2018) inventory compared to the LUBW inventory. In the EDGAR v6.0 (2014-2018) inventory, $41\%$ of anthropogenic $CH_4$ is emitted from the natural gas sector and

$9\%$ from waste incineration. The LUBW inventory reports only $28\%$ of anthropogenic $CH_4$ from the natural gas sector and does not include emissions from waste incineration.

## 3.6 Comparison between mean $\delta^{13}$C isotopic source signatures calculated with atmospheric measurements and emission inventories

The mean $\delta^{13}$C isotopic source signatures calculated for the LUBW and EDGAR v6.0 inventories are compared to the mean

isotopic source signature determined out of atmospheric measurements. Figure 9 shows the mean $\delta^{13}$C isotopic source signatures for each month and the annual averages (dashed lines), which are determined out of atmospheric measurements (black) or using the EDGAR v6.0 inventory (blue). The dashed red line displays the annual mean source signature calculated with the LUBW inventory. The monthly values shown as solid red line are determined using the annual LUBW data, including a modelled annual cycle for the sector small and medium-sized combustion plants (KuMF). This modelled annual cycle is based

on the most prominent annual cycle in the $CH_4$ emissions estimated by EDGAR v6.0: the one for the energy for building sector.

The annual mean $\delta^{13}$C isotopic signatures determined using EDGAR v6.0 are approximately $7‰$ more enriched than the results from atmospheric measurements calculated with the moving Miller-Tans approach. The results from the LUBW inventory show similar values to the mean source signatures determined out of atmospheric measurements, with only a small difference of less than $1‰$. Furthermore, mean source signatures calculated using the two inventories also show an annual cycle with

more depleted values in summer. However, the peak-to-peak amplitude in the annual cycle determined out of atmospheric



measurements is 6.2‰ and thus approximately three times larger than the annual cycles noticeable by EDGAR v6.0 and the modelled LUBW data. Thus, the observed annual cycle resulting from atmospheric measurements can only be partly explained by seasonal variations of $CH_4$ emissions from heating. This indicates that emissions from another sector have relevant seasonal variations too, which are not yet included into EDGARv6.0 inventory.

By using inverse models, Bergamaschi et al. (2018) found an annual cycle in $CH_4$ emissions in Germany, with the maximum in summer. Due to the limited number of studies, they could not quantitatively estimate potential seasonal variations of anthropogenic sources (Bergamaschi et al., 2018). However, some studies such as Ulyatt et al. (2010), Spokas et al. (2011) and VanderZaag et al. (2014) reported an annual cycle in $CH_4$ emissions from biogenic sources such as dairy cows, landfills or waste water with more emissions in summer. Such seasonal variations in biogenic emissions, in addition to the variations

of emissions from heating, can explain the annual cycle in the catchment area of Heidelberg determined by atmospheric measurements.

## 4    Conclusion

In this study, the continuous time series of atmospheric $CH_4$ and $\delta(^{13}CH_4)$ measured over six years in Heidelberg is used to study seasonal variations and trends of $CH_4$ emissions in the catchment area of Heidelberg. The $CH_4$ mole fraction increases

by $(6.8 \pm 0.3)\,\mathrm{nmol\,mol^{-1}\,a^{-1}}$ between 2014 and 2020 and $\delta(^{13}CH_4)$ shows a decreasing trend of $(-0.028 \pm 0.002)\,‰\,a^{-1}$. Furthermore, $CH_4$ and $\delta(^{13}CH_4)$ show strong annual cycles with the minimum in late summer and early autumn, respectively.

The partitioning of local and regional $CH_4$ emissions among different source categories is analysed by determining the mean $\delta^{13}C$ isotopic source signature in the catchment area of Heidelberg. Therefore, the Miller-Tans method in combination with the York fit are applied to the measured atmospheric $CH_4$ and $\delta(^{13}CH_4)$ time series. Three different approaches are tested which

correspond to different time intervals: the monthly approach, the night-time approach and the moving Miller-Tans approach. In all these approaches, no significant trend in the monthly mean source signature occurs during the last six years. This confirms that the source composition in the catchment area of Heidelberg did not change between 2014 and 2020.

The average mean source signatures calculated with the above described approaches vary between $-52.3‰$ and $-53.9‰$. The $CH_4$ emissions measured in Heidelberg originate from different sources in the urban area as well as in the rural surround-

ings. They range from biogenic sources, such as livestock over waste treatment, to thermogenic sources, such as natural gas, and even to pyrogenic ones, such as traffic and wood-firing installations. The determined monthly mean $\delta^{13}C$ isotopic source signatures of all approaches show an annual cycle with a peak-to-peak amplitude of $6.2‰$ and a stronger biogenic $CH_4$ contribution in summer. The comparison with emission inventories have shown that this cycle can only be partly explained by seasonal variations in the $CH_4$ emissions from heating. Thus, additional seasonal variations probably occur in biogenic $CH_4$

emissions from waste water, landfills or dairy cows. However, there is still a great need for research in order to understand and describe potential annual cycles of $CH_4$ sources precisely.

Furthermore, the mean $\delta^{13}C$ isotopic source signatures determined for the catchment area of Heidelberg using atmospheric measurements are used to verify the $CH_4$ emissions reported by two emission inventories. EDGAR v6.0 seems to overestimate



$CH_4$ emissions from more enriched sources. The mean source signature resulting from EDGAR v6.0 data is around 7‰ more enriched than the one determined from atmospheric measurements. This large difference can be partly explained by the large amount of $CH_4$ emissions estimated by EDGAR v6.0 for waste incineration and the energy for buildings sector. The LUBW inventory estimates much lower $CH_4$ emissions than EDGAR v6.0, especially for the waste sector. The mean $\delta^{13}C$ isotopic source signature calculated using the emissions reported by LUBW agrees well with the result from atmospheric measurements. This study gives a first impression about how well the emission inventories represent the $CH_4$ emissions in the catchment area of Heidelberg.

*Data availability.* The $CH_4$ mole fraction and $\delta(^{13}CH_4)$ time series from Heidelberg are available at doi:10.11588/data/OXKVW2 and on request from the data owner (martina.schmidt@iup.uni-heidelberg.de).

*Author contributions.* AH and MS designed the study and were responsible for the $CH_4$ mole fraction and $\delta(^{13}CH_4)$ measurements in Heidelberg. AH evaluated the data and wrote the paper with the help of MS.

*Competing interests.* The authors declare that they have no conflict of interest.

*Acknowledgements.* The authors would like to thank Michael Sabasch and Ingeborg Levin from the Institute of Environmental Physics in Heidelberg as well as the Stable Isotope Laboratory at Max Planck Institute for Biogeochemistry (MPI-BGC) in Jena for the calibration of standard gases. We also thank Frank Meinhardt for his great support during the measuring campaigns at Schauinsland station operated by the German Environment Agency (UBA). The measuring campaigns at Schauinsland and their data analysis were funded by Projects 95297 and 167847 with the German Environment Agency (UBA). We wish to thank Henrik Eckhardt and Julia Wietzel for their great support in this study, and William Cranton for proofreading this article. The CRDS analyser was funded through the DFG excellence initiative II.



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





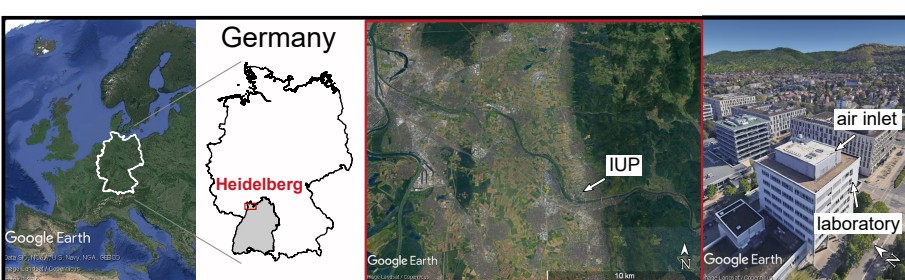

**Figure 1.** Location of the measurement station in Heidelberg at the Institute of Environmental Physics (IUP - Institut für Umweltphysik) (map data on from © Google Earth).





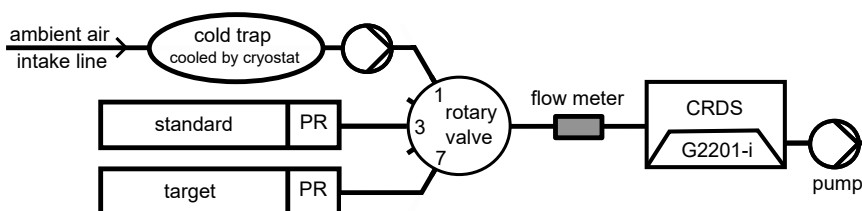

**Figure 2.** Experimental setup to measure $CH_4$ mole fraction and $\delta(^{13}CH_4)$ in ambient air in Heidelberg.



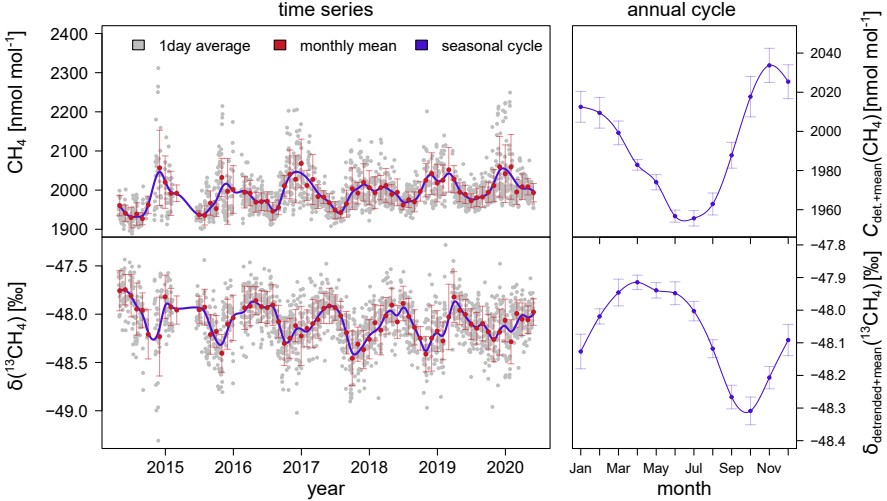

**Figure 3.** Atmospheric CH$_4$ mole fraction and $\delta(^{13}\mathrm{CH}_4)$ measured in Heidelberg and corresponding annual cycles. The monthly mean values and standard deviation (red) are calculated from the daily averages (grey). The mean annual cycle with the standard errors are shown in blue.





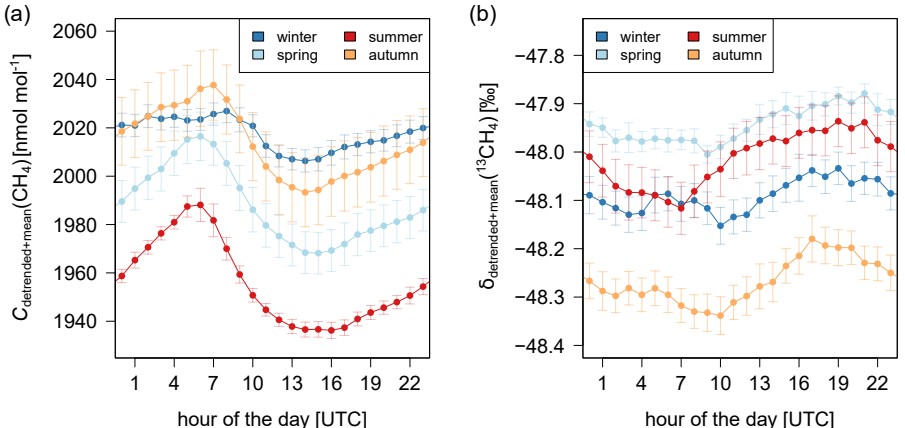

**Figure 4.** Diurnal cycles of $CH_4$ (a) and $\delta(^{13}CH_4)$ (b) in Heidelberg. For each season the diurnal cycles of each month, which are detrended by subtracting the diurnal mean, are averaged and the mean $CH_4$ mole fraction or $\delta(^{13}CH_4)$ value for each season is added.





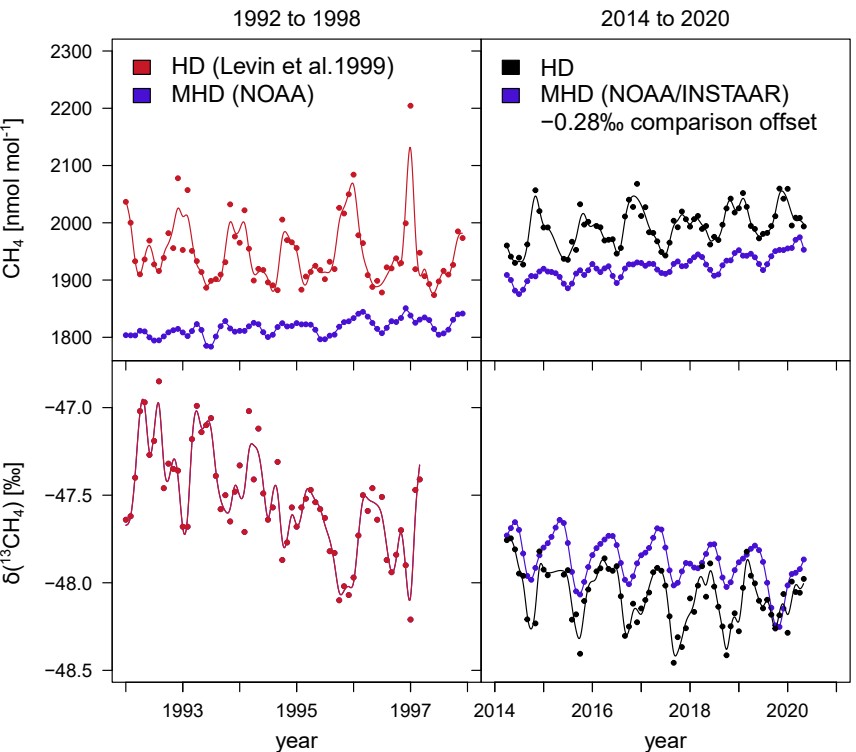

**Figure 5.** CH$_4$ mole fraction and $\delta(^{13}\text{CH}_4)$ in Heidelberg from 1992 to 1998 (Levin et al., 1999) and between 2014 and 2021. In addition, measurements done at the marine background station Mace Head (Lan et al., 2022; Michel et al., 2022) are shown in blue.





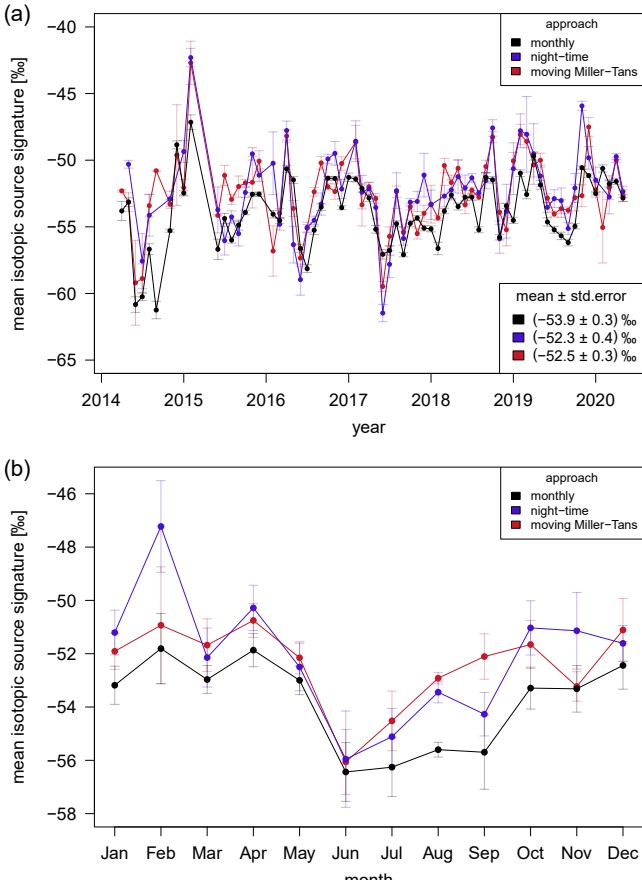

**Figure 6.** The monthly averages (a) and the annual cycle (b) of the mean $\delta^{13}$C isotopic source signatures of CH$_4$ in the catchment area of Heidelberg between April 2014 and May 2020. The monthly (black), the night-time (blue) and the moving Miller-Tans (red) approach are used for the determination. The error bars corresponds to the standards deviations.





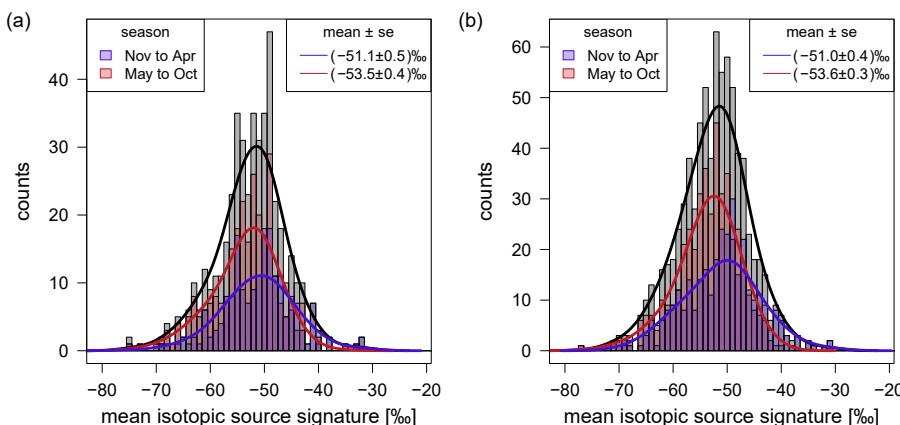

**Figure 7.** Frequency distribution of the determined mean CH$_4$ isotopic source signatures of individual nights (a) or events (b) in the catchment area of Heidelberg.

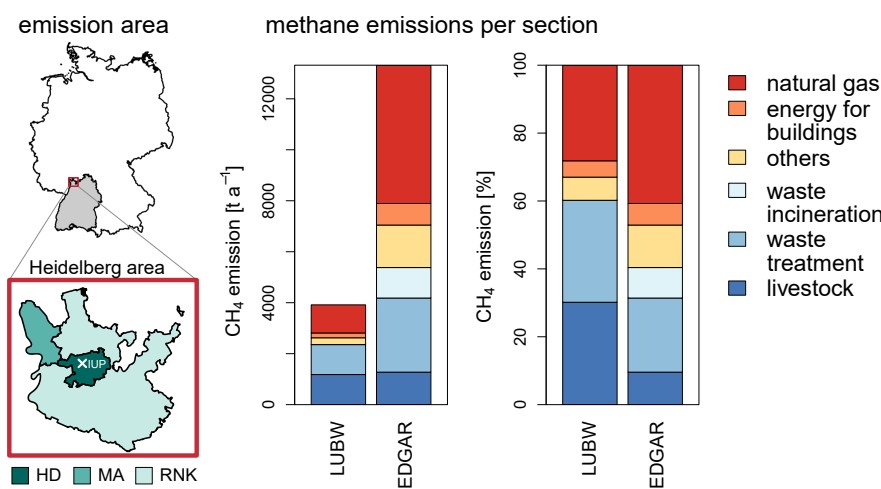

**Figure 8.** CH$_4$ emissions and relative proportion of different source categories reported by LUBW (LUBW, 2016) and calculated from EDGAR v6.0 (Crippa et al., 2021) data for the Heidelberg area, which includes the cities of Heidelberg (HD) and Mannheim (MA) as well as the county Rhein-Neckar-Kreis (RNK).





**Table 1.** Isotopic $^{13}$C signatures of different CH$_4$ sources based on measured values in the catchment area of Heidelberg and literature: (1) Hoheisel et al., 2019, (2) Levin et al., 1993, (3) Sherwood et al., 2017, (4) Widory et al., 2006 (for $\delta(^{13}CO_2)$), (5) Menoud et al., 2021 and (6) Zazzeri et al., 2017.

| Sector | Source | Isotopic $^{13}$C signature [‰] |
|---|---|---|
| livestock farming | ruminants[1] | $-63.9 \pm 1.3$ |
| solid waste landfills | landfill[1] | $-58.7 \pm 3.3$ |
| waste water treatment | waste water treatment plant[1] | $-52.5 \pm 1.4$ |
| exploitation of oil and coal | coal from Europe and Russia[3] | $-46.6 \pm 6.4$ |
| gas distribution | natural gas[1] | $-43.3 \pm 0.8$ |
| waste incineration | waste incineration[4] | $-33.2 \pm 4.6$ |
| energy for buildings | non-industrial combustion[5] | $-31.1$ |
| industrial emissions | combustion (industrial)[6] | $-25$ |
| road transport | cars[2] | $-22.8$ |





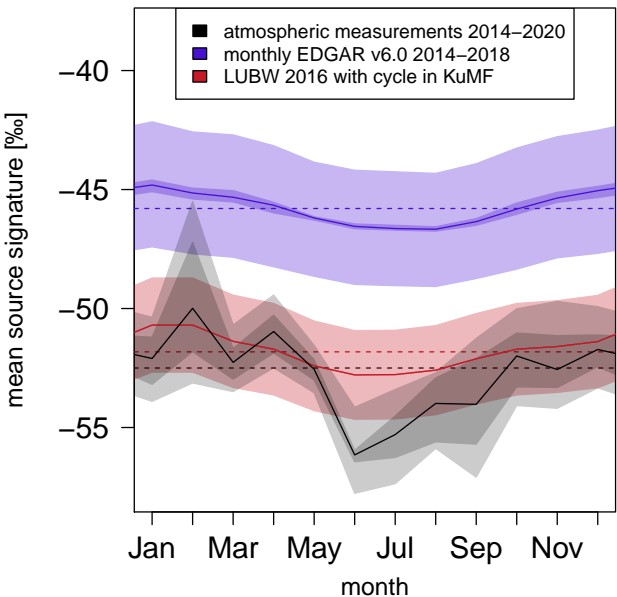

**Figure 9.** Annual variability in the monthly mean $CH_4$ isotopic source signatures calculated with emission inventories and atmospheric measurements for the Heidelberg area. The light blue and red areas for EDGAR v6.0 (Crippa et al., 2021) and LUBW (LUBW, 2016) corresponds to errors in the applied source signatures and the dark blue area for EDGAR v6.0 shows differences in the $CH_4$ isotopic source signatures for all years between 2014 and 2018. The dark grey area corresponds to the results for atmospheric measurements from the different approaches and the light grey area includes errors, too.



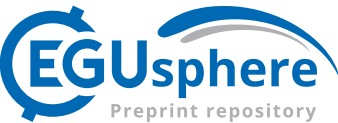

**Table A1.** CH$_4$ mole fraction and isotopic ratio of the two calibration gases used to calibrate the ambient air measurements carried out in Heidelberg.

| period of use | CH$_4$ [nmol mol$^{-1}$] | $\delta(^{13}$CH$_4)$ [‰] |
|---|---|---|
| up to August 2019 | $1934.5 \pm 0.1$ | $-47.83 \pm 0.05$ |
| from August 2019 | $2003.6 \pm 0.4$ | $-48.10 \pm 0.07$ |



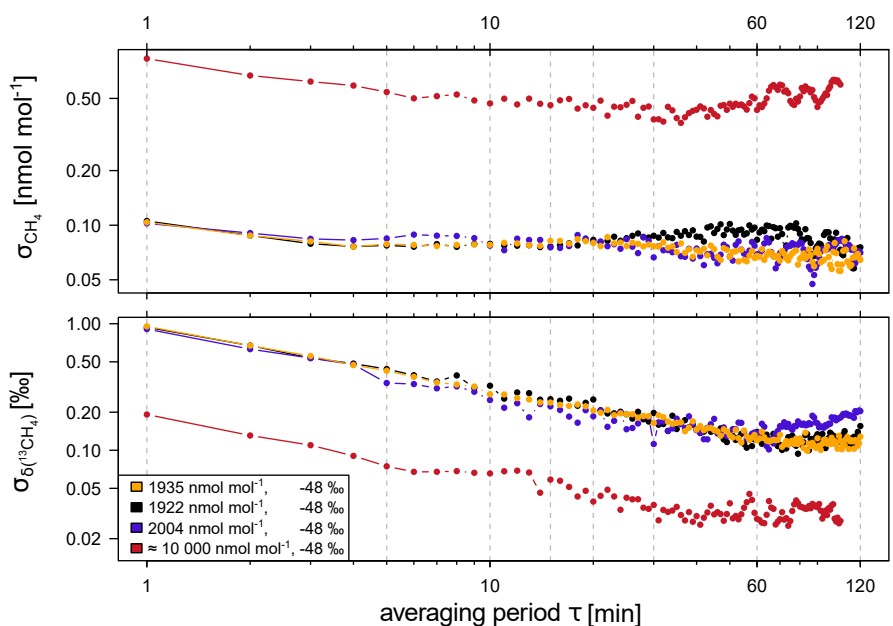

**Figure A1.** Allan standard deviations for CH$_4$ mole fraction and $\delta(^{13}$CH$_4)$ determined for the CRDS G2201-i analyser and different CH$_4$ mole fractions and isotope ratios. The Allan standard deviations are based on measurements from 2013 (orange) and 2019 (black, blue, red).





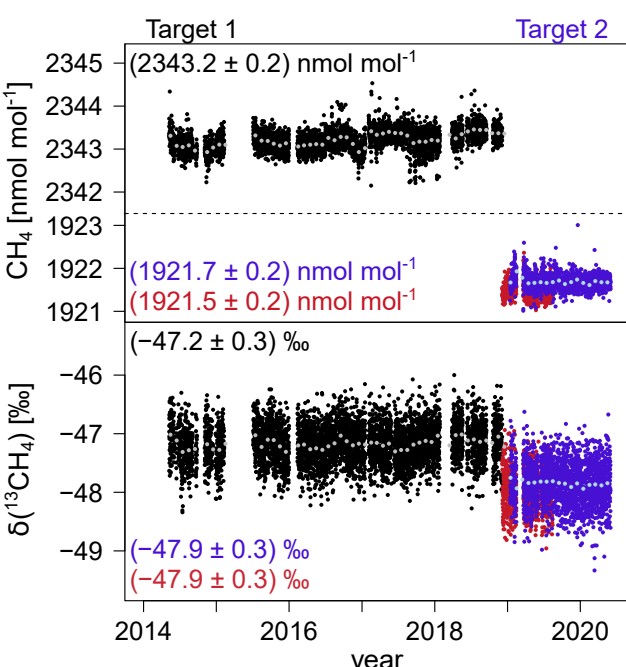

**Figure A2.** Calibrated $CH_4$ mole fractions and $\delta(^{13}CH_4)$ values of the target cylinder measurements. Target 1 (calibrated with calibration cylinder 1) is shown in black and Target 2 (calibrated with calibration cylinder 2) in blue. The grey and light blue data points correspond to the monthly average values. For quality control, Target 2 was additionally calibrated with calibration cylinder 1 and is shown here in red.





**Table A2.** $\delta(^{13}\text{CH}_4)$ measurements of six intercomparison cylinders. The $\delta(^{13}\text{CH}_4)$ values determined by MPI-BGC are taken from Umezawa et al. (2018) and are compared with our results. The difference of the multiple measurements is shown in parenthesis and the uncertainty of the average difference is given as the standard error of the mean.

| sample ID (collection date) | analysis date MPI-BGC | analysis date UHEI-Pic | $\delta(^{13}\text{CH}_4)$ MPI-BGC [‰] | $\delta(^{13}\text{CH}_4)$ UHEI-Pic [‰] | difference UHEI−MPI [‰] |
|---|---|---|---|---|---|
| GvN 88/20 (Jul 1988) | Jul 2013 | May 2018 & May 2019 | −47.66 (0.07, N= 2) | −47.60 (0.29, N= 3) | +0.06 |
| GvN 92/12 (May 1992) | Jun 2013 | May 2018 & May 2019 | −47.40 (0.04, N= 2) | −47.61 (0.19, N= 4) | −0.21 |
| GvN 96/03 (Feb 1996) | Jun 2013 | May 2018 & Apr 2019 | −47.18 (0.26, N= 2) | −47.07 (0.23, N= 3) | +0.11 |
| GvN 99/14 (Dec 1999) | Jul 2013 | Jun 2018 & Apr 2019 | −47.23 (0.16, N= 2) | −47.13 (0.02, N= 2) | +0.10 |
| GvN 06/14 (Sep 2006) | Jul 2013 | May 2019 & Feb 2020 | −47.19 (0.09, N= 2) | −47.26 (0.23, N= 3) | −0.07 |
| GvN 08/03 (Mar 2008) | Jun 2013 | Feb 2020 | −47.35 (0.05, N= 2) | −47.24 (0.37, N= 2) | +0.11 |
| average | | | | | $(+0.02 \pm 0.05)$‰ |