# Peer review of "Six years of continuous carbon isotope composition measurements of methane in Heidelberg (Germany) — a study of source contributions and comparison to emission inventories"

_EGUsphere, 2023_

## Referee Comment (RC2)

In this paper, Hoheisel et Schmidt describe new continuous CH4 and δ(13C,CH4) measurements retrieved between 2014 and 2020 In Heidelberg (Germany). After introducing the experimental setup, they analyze the temporal variability of this data and apply the Miller-Tans method to derive estimates of the mean isotopic signature that could cause these variations. These determined estimates are then compared to bottom-up estimates using two different inventories.

Overall, the paper is well presented and well written. The structure is clear and it is easy to understand where the authors are leading us. Also, the scientific questions addressed in this study are well within the scope of ACP and the analysis conducted to answer these questions is detailed, elaborate and tackles very interesting points, both for experimentalists and atmospheric modelers. Last but not least, this new continuous data is invaluable to better investigate methane sources and will likely be utilized in the future by the rest of the atmospheric community.

Most of my comments only call for additional clarity in the methodology and the presentation of results. Also, a few additional details in the methodology and in the results would be beneficial both for the reproducibility of the study and the comprehensiveness of the analysis. However, these comments are very minor and I can already recommend this paper for a publication in the journal ACP.

**Specific comments**

Line 1: I recommend not using the abbreviation δ(13CH4) in the abstract. Use δ(13C,CH4).

Line 2: You write that it is a 6-year time-series, since 2014. It may give the reader the impression that the measurements stopped in 2020. I suggest a small revision: "Between 2014 and 2020, the time series shows an increasing trend of (6.8 ± 0.3) nmolmol−1 a−1 for the CH4 mole fraction."

Line 6: At present, it seems you are using δ(13C,CH4) (abbreviated as δ(13CH4)) for atmospheric isotopic composition and δ13C (also an abbreviation of δ(13C,CH4)) for isotopic signature. In my opinion, it's okay to keep it that way but you should not use the abbreviations in the abstract and also introduce the abbreviation δ13C in the main text.

Line 9: Sentence not clear. As far as I understand, the mean estimated δ(13C,CH4) source isotopic signature exhibits a seasonal variation, with a peak-to-peak variation -6.2 ‰. If it is the case, you should reformulate. I suggest replacing the sentence "This annual cycle in 13C-CH4 sources…", with "This annual cycle in the mean source isotopic signature source, with a peak-to-peak amplitude of −6.2‰, can only be partially explained by seasonal variations in the 13C-enriched emissions from heating."

Line 25: After this sentence, you should introduce the isotopic scale δ(13C,CH4) with a formula. Because after that you provide typical source signature values for different source categories (e.g. −55‰ to −70‰) , but the reader does not know what scale you are referring to. You could also be talking about 14C rather than 13C.

Line 30: Give the recent references for these values also here (e.g., Sherwood et al. 2017; 2021; Menoud et al., 2022). Also, the thermogenic range you provide appears slightly inconsistent (too small and too enriched) when compared to the information presented in these references.

Line 59: "To our knowledge, our time series is the longest in situ δ(13CH4) record, with high temporal resolution, reported to date.". You mean the longest for Heidelberg or the longest ever in the world ?

Line 64: A continuous six-year time series between when and when ?

Line 99: Would be worth mentioning the value you are using for the reference 13C/12C ratio because there is sometimes confusion between PDB and VPDB values.

Line 119: I think two significant figures is not enough for δ(13CH4). I would recommend three, as you do in the rest of the paper. If we look at the bulk, I mostly see values between -49.0‰ and -47.5‰. If you also want to include extreme values, then it is approximately -49.5‰ to -47.2‰

Line 123: Please plot the trend on Figure 3. I think if you increase the size of the figure and increase the transparency of the 1day averages, then it won't decrease the overall clarity.

Line 127: Less enriched compared to what ? To the mean ? Overall, I think some confusion can arise from the fact that you use the mean over the full time series as the center of your annual cycle, rather than using zero. Line 128 and 129 suggest that the values are always -48.3‰ in early autumn and -47.9‰ in spring, while these values have an interannual variability, due to the trend and the variations in the seasonal cycle. It's okay to keep it that way but you should comment on that.

Line 139: After the analysis in Section 3.3, can you think of a reasonable explanation ?

Line 146: Could you provide the details of the Mace Head Observatory (altitude, longitude, latitude) ? Also, please explain why Mace Head can be considered as a "background" station.

Line 156: In this paragraph, you use both "emissions in Heidelberg" and "emissions in the catchment area of Heidelberg". Is it supposed to mean the same thing (I suppose so) ? Or did Levin et al. (2011, 2021) only analyze emissions in the city of Heidelberg (without the surroundings) ?

Line 166: You should very briefly introduce the abbreviation δ13C the same way you introduced it for δ(13CH4) for the atmospheric isotopic composition.

Line 182: If you want to make a comparison between the Miller-Tans method and the Keeling plot, which is a good idea, you should briefly introduce the Keeling plot method as well.

Line 187 and line 200: Which CH4 range ? Apologies for being confused here. You discard every data point where the difference between $C_{bg}$ and $C_{obs}$ is below 100 nmol mol$^{-1}$ ? If it is the case, for the night data set, do you discard the full night if one of the data points does not satisfy this criteria ?

Line 193: Please, reformulate. What can be assumed ? That it is constant ?

Line 213: What do you mean by 'directly adjacent in time' ? What amount of time do you consider to be 'adjacent in time' ? And what percentage of the 18% are during night time ?

Line 222: Here and throughout the text, you often use "more/less depleted" or "more/less enriched". Usually, a value is depleted/enriched compared to a point of comparison, which is often the atmospheric value (around -48.07‰ in your case), as you state it very clearly in the following sentence. For instance, in this situation, I would use "more depleted" rather than "less enriched" because the source signatures for the moving Miller-Tans and night time approaches are already depleted compared to the atmospheric composition. Therefore, the third one is even more depleted.

Line 224: It seems that you are suggesting there is a causality between the fact that the estimated source isotopic signature is more depleted than the atmospheric composition and the fact that biogenic sources play a dominant role. I do not think it is true. You can suggest biogenic sources are dominant because the estimated source signature is low and close to what could be expected if biogenic sources (typically between -55‰ and -70‰) were dominant. Or to the contrary, it would be too low compared to a situation where only pyrogenic and thermogenic were dominant (although some thermogenic sources can have a source isotopic signature as low as -60‰, see Sherwood et al., 2017).

Line 271: Please reformulate. Do you mean both explanations are plausible ? Is it exclusive ? The too at the end is a bit misleading. In general, I do not understand how long an "event" is. Therefore, it is difficult to confirm that a night time increase can influence the event detected by the Miller-Tans approach. It would be nice to show somewhere the typical length of an event. Do these events happen mostly during the night ? As far as I understand, you can access this information with your methodology.

Line 280: You suggest two explanations but as far as I understand, both explanations are closely linked. Small pollution events of the first explanation can be the ones from distant sources from the second explanation. If it is correct, maybe you could mention it at the end of the paragraph.

Line 292 and onwards: Again "more depleted" compared to what ?

Line 303: "The monthly values vary on average between 0.1‰ and 0.8‰". What does the percentage represent ?

Line 319: Give the exact location of the station.

Line 340: You have only one subsection 3.4.1 under section 3.4. Shouldn't section 3.5 and 3.6 be subsections of sections 3.4 ? Or at least section 3.5 ?

Line 354: Why does it "seem" to decrease ? If you are not confident, where does this number 7%, without any uncertainty, come from ?

Line 389: It is not clear what you needed.

**Technical comments**

Figures: Although they have a good resolution, I would have preferred the figures to be larger, i.e. fitting the width of the page.

Figure 1: In the caption, map data on from → map data from

Figure 3: There is a problem with the x-axis ticks of the top-right panel. Please, make it similar to the bottom-right panel. Also, please add in the caption a note saying that the y-axis ranges are not the same for the left and right panels.

Line 67: to CH4 **total** emissions.

Line 152: Replace with "This was different" or "this is different for the 1990s"

Line 180: Put these subscripts in the same format as in the equation.

Line 214: Do you mean 2014 instead of 2011 ?

Line 266: Although → However

Line 274: less enriched → more depleted

Line 294: remove the space between the first parenthesis and "see"

Line 331: go one step further → extend the effort ?

Line 378: This is supported by the fact that the amount of emissions from sectors, such as livestock farming, with well studied emission factors and accurate statistical data are comparable for both inventories → This is supported by the fact that the amount of emissions from sectors with well studied emission factors and accurate statistical data are comparable for both inventories, such as livestock farming, are comparable for both inventories.

**References**

Sherwood, O. A., Schwietzke, S., Arling, V. A., and Etiope, G.: Global Inventory of Gas Geochemistry Data from Fossil Fuel, Microbial and Burning Sources, version 2017, Earth Syst. Sci. Data, 9, 639–656, https://doi.org/10.5194/essd-9-639-2017, 2017

Menoud, M., van der Veen, C., Lowry, D., Fernandez, J. M., Bakkaloglu, S., France, J. L., Fisher, R. E., Maazallahi, H., Stanisavljević, M., Nęcki, J., Vinkovic, K., Łakomiec, P., Rinne, J., Korbeń, P., Schmidt, M., Defratyka, S., Yver-Kwok, C., Andersen, T., Chen, H., and Röckmann, T.: New contributions of measurements in Europe to the global inventory of the stable isotopic composition of methane, Earth Syst. Sci. Data, 14, 4365–4386, https://doi.org/10.5194/essd-14-4365-2022, 2022.

---

## Author Comment (AC1)

**Author's Response to Referee#1 Comments on:** "Six years of continuous carbon isotope composition measurements of methane in Heidelberg (Germany) – a study of source contributions and comparison to emission inventories"

We thank the Referee#1 for the detailed and constructive comments and their useful suggestions. We have revised the manuscript accordingly.

**General Comments:**

[Referee#1]    The paper "Six years of continuous carbon isotope composition measurements of methane in Heidelberg (Germany) – a study of source contributions and comparison to emission inventories " is a detailed analysis of observed methane mole fraction and carbon isotope signature in Heidelberg, supported by elaborated discussion of possible origin of observed methane elevation and comparison with existing inventories. The paper focus on observed trend in methane mole fraction and carbon isotopic signature over 6 years and the title clearly reflect the contents of the paper. Also, an abstract provides a concise and complete summary. Overall presentation is well organised and deliberated, and the used language is fluent and precise, making the paper easy to follow and understand. The paper can be treated as case study of methane long-term observation in the urban area. Overall, the paper address relevant scientific questions within the scope of ACP and presents novel data with its interpretation, which are useful to the atmospheric community. The substantial conclusions are reached, showing the long-term trend and similarities and discrepancies with other atmospheric studies and inventories. The scientific method and assumptions are clearly outlined. The specification of measurement site and used instrument (Allan deviation, long-term reproducibility, accuracy based on comparison with MPI-BGC measurements) is well described.

However, there is not too much explanation of used Miller-Tans method, especially there is no information about extracted background and its potential impact for determined δ13C signature of methane source. More elaborated description of comparison between Keeling and Miller-Tans would be also useful. Also, giving more details about implementation of Miller-Tans method will make results more traceable and reproductive.

Overall, the paper is well balanced, clear and, containing appropriate references and gives important contribution to atmospheric studies of methane. Some questions and comments should be taken into consideration before publishing.

**Specific comments:**

[Referee#1]    The more detailed explication of used Miller-Tans method in this study is missing. What was background used in Miller-Tans method? How the background was chosen and how the choice of background could affect obtained values using Miller-Tans method?
Line 182: What tests was made to compare Keeling and Miller-Tans methods? Also, could you elaborate more about the fact you did not observe differences between Keeling and Miller-Tans method? There are studies showing that the differences are observed using these two different methods.

[Hoheisel and Schmidt]    Thank you for your detailed comments and questions regarding the Miller-Tans and Keeling plot methods. Your comments have shown us that we were not clear in the manuscript and that the term "Miller-Tans method" can lead to confusion.
In our study, we used equation 5 derived from Miller and Tans (2003):
$$\delta_{obs}\, C_{obs} = \delta_s\, C_{obs} + C_{bg}\, (\delta_{bg} - \delta_s)$$

In this form of the 'Miller-Tans method', the background values can remain unknown and must not be specified. This is different in equation 6 from Miller and Tans (2003), which we did not used.

In addition to the 'Miller-Tans method (equation 5)', we also used the Keeling plot method to determine the mean isotope source signatures. A comparison between the mean isotope source signatures calculated with the 'Miller-Tans method (equation 5)' and the Keeling plot method using the York fit (York et al., 2004) showed no significant difference. All our results are identical in more decimal places than the significant digits.

Since we tested both the 'Miller-Tans method (equation 5)' and the Keeling plot method in our study, and all results are identical regardless of which of the two methods was used, we have decided to explain our results using the Keeling plot method rather than the 'Miller-Tans method'. We have made the necessary changes in the revised manuscript and hope that the used methods are now clearer and the results of our study more traceable.

| [Referee#1] | Why KuMF results were included in section 3.6 but not in section 3.5? How monthly $\delta^{13}$C-CH$_4$ signatures from inventories were calculated? |
|---|---|
| [Hoheisel and Schmidt] | Emissions from KuMF are always included in the LUBW inventory. Emissions from the same sources have different sector names in the EDGAR and the LUBW inventories. The 'small and medium-sized combustion plants (KuMF)' sector reported by LUBW and the 'energy for buildings' sector from EDGAR describe the same emission category. Therefore, in accordance with the EDGAR inventory, the KuMF sector was referred to as 'energy for buildings' when comparing both inventories. When describing the preparation of the artificial monthly LUBW data, we have used the term KuMF to make it clearer which LUBW sector we have used. We understand, that this can lead to confusion. Therefore, we have changed the paragraph in the revised manuscript. To determine the monthly mean $\delta^{13}$C-CH$_4$ isotopic source signatures, we assign a source-specific isotopic signature to the monthly emissions from each sector. EDGAR already reports monthly CH$_4$ emissions. Since LUBW only reports annual emissions, we determined monthly values by dividing the annual values for each sector by 12. We then included an annual cycle in the emissions of the energy for buildings (KuMF) sector, analogous to the annual cycle reported by EDGAR. |

*"EDGARv6.0 reports monthly CH$_4$ emissions, which were used to calculate the monthly mean isotopic carbon source signatures. The most prominent annual cycle in the CH$_4$ emissions estimated by EDGARv6.0 occurs in the energy for buildings sector. The LUBW only reports annual emissions. Therefore, we included a modelled annual cycle for the energy for buildings sector (the LUBW sector small and medium-sized combustion plants - KuMF). This modelled annual cycle is based on the annual cycle noticeable in the monthly EDGARv6.0 emissions for the energy for buildings sector."*

| [Referee#1] | Discussion about discrepancies between measurements and inventories in other cities, including comparison with Heidelberg is worth to add. |
|---|---|
| [Hoheisel and Schmidt] | Thank you for the comment. We agree that it is a valuable contribution to our analysis, to include results of other studies in cities, which also analyse the discrepancies between measurements and inventories. We have included this in the revised manuscript. |

| [Referee#1] | Line 30-35 Given range of microbial and thermogenic is narrower than in the literature (e.g. Menoud et al, 2022) and not overlapping as it is observed during source signature studies. Please clarify. |
|---|---|
| [Hoheisel and Schmidt] | This paragraph in the introduction is intended to give an overview of the different $CH_4$ sources and their typical isotopic source signatures. As indicated in the manuscript, we have taken the typical isotopic source signatures from the IPCC AR5 WG1 report from 2013. Individual measurements and especially more recent measurements may of course deviate from these typical values.  To avoid any misunderstandings, we have changed the paragraph and added a sentence that discusses the more recent measurement results. |

"*$CH_4$ is emitted from anthropogenic and natural sources, which are grouped in three different categories according to the production processes. Biogenic $CH_4$ is produced under anaerobic conditions due to degradation of organic matter (typically -70‰ to -55‰; IPCC, 2013). Biogenic $CH_4$ sources are wetlands, ruminants, landfills and wastewater treatment plants. Thermogenic $CH_4$, like that in natural gas, is formed on geological time scales out of organic matter and is less depleted than biogenic $CH_4$ (typically -45‰ to -25‰; IPCC, 2013). Pyrogenic $CH_4$ is formed during the incomplete combustion of organic matter, such as biomass burning, and is more enriched (typically -25‰ to -13‰; IPCC, 2013) compared to biogenic and thermogenic $CH_4$. Studies by Sherwood et al. (2017; 2021) and Menoud et al. (2022) show that the $\delta^{13}C$-$CH_4$ values of the different source categories are not always as distinct as indicated above. They give much larger ranges of $\delta^{13}C$-$CH_4$ values for the different source categories, which also overlap as a result. Especially for fossil but also for biogenic sources large regional differences occur.*"

| [Referee#1] | Line 42- 45: It would be also worthy to include and cite paper of Rennick et al 21 (https://doi.org/10.1021/acs.analchem.1c01103) as it is another laser spectrometry method for methane isotopes measurements. |
|---|---|
| [Hoheisel and Schmidt] | We have included the study in the revised manuscript. |

| [Referee#1] | Line 121 Could you add short description (e.g., one sentence) to explain principal of CCGCRV? |
|---|---|
| [Hoheisel and Schmidt] | Thank you for pointing this out. We have added a short description of CCGCRV in the revised manuscript: |

"*CCGCRV can be used to decompose a time series into a trend and a detrended seasonal cycle by fitting a polynomial equation combined with a harmonic function to the data and applying a filter to the residuals. In this study, we used 3 polynomial terms and 4 annual harmonic terms. The short- and long-term cutoff values for the low-pass filter are 80 and 667, respectively.*"

| [Referee#1] | Line 146: What is frequency of used Mace Head data? What is the height of the inlet in Mace Head? Why Mace Head was used? |
|---|---|
| [Hoheisel and Schmidt] | We used monthly mean Mace Head data and included more information about the Mace Head Observatory in the revised manuscript: |

"*The Mace Head Observatory (53°19'36"N, 9°54'16"E, 8.4m a.s.l.) is located on the west-coast of Ireland and measures the maritime background mole fraction when air is coming from the ocean.*"

We are aware that the Mace Head Observatory is not optimal as a background station for Heidelberg. However, since this is the only background station west of Heidelberg with a long published $\delta^{13}C$-$CH_4$ record, we have decided to use the data from Mace Head to characterise and compare the Heidelberg measurements of today (2014-2020) and the 1990s.

| | |
|---|---|
| [Referee#1] | Line 177: Why Allan standard deviation was used as uncertainty instead of standard deviation? |
| [Hoheisel and Schmidt] | In the Miller-Tans/Keeling plot approach, we perform a York fit that includes the uncertainty in x and y. To determine these uncertainties, we used the Allan standard deviation instead of the standard deviation to account for instrumental uncertainty rather than atmospheric variability included in the averaged value. |

| | |
|---|---|
| [Referee#1] | Line 209: Why the method to extend for another hour, up to 12 hours was chosen? Why 12 hours was chosen as criteria to exclude data? |
| [Hoheisel and Schmidt] | For the moving Miller-Tans/Keeling plot approach, we tested two scenarios: We started with 1 hour and increased the interval by hourly steps if our criteria were not met, and additionally, we started with a 12-hour time window and decreased it by hourly steps if our criteria were not fulfilled. We noticed no significant difference between the monthly averaged mean isotopic source signatures calculated from the two scenarios. Since we are interested in short-term events, we presented the scenario where we extended the time window. |

| | |
|---|---|
| [Referee#1] | Line 205-214: The one minute step seems quite small, especially that some pollution peaks can last longer. You mentioned you averaged all values directly adjacent in time. Was it done manually? Is it enough valid method to separate individual pollution event? Would wider step (e.g., few minutes) be more adequate? |
| [Hoheisel and Schmidt] | We automatically applied the moving Miller-Tans/Keeling plot approach to the complete time series of six years. For each time, we chose the smallest time interval for the moving Miller-Tans/Keeling plot, which fulfilled our criteria. Therefore, we achieve results for 18% of the one-minute averaged data. Based on your comments and the ones from Referee#2, we had a closer look at the automatically generated events and decided not to split the moving Miller-Tans/Keeling plot results into individual events any more. Instead, we examine the hourly and daily averages of the results of the moving Keeling plot. We changed this in the revised manuscript accordingly. |

| | |
|---|---|
| [Referee#1] | Line 235-245: First you say there is no significant trend in the monthly mean $\delta^{13}C$ isotopic signatures, while later you describe visible differences between signatures for individual months. Please clarify. |
| [Hoheisel and Schmidt] | As mentioned in the manuscript, for each approach individually, there is no significant trend detectable in the monthly mean isotopic carbon source signatures between 2014 and 2020. However, when comparing the monthly mean isotopic carbon source signatures determined from different approaches, we can notice differences in individual month and in the annual cycle. |

| | |
|---|---|
| [Referee#1] | Line 262-267: Could choosing the wider step than 1 minute could remove possible artefact of averaging and give more reliable values to determine diurnal cycle? |
| [Hoheisel and Schmidt] | For each time x, the mean isotopic source signature is determined by applying the Miller-Tans/Keeling plot approach to the minutely-values in a 1 to 12 hours interval around the time x. Choosing a wider step of several minutes instead of one minute would only decrease the number of resulting mean isotopic source signature values, but should not change the monthly averages significantly. Furthermore, we also tested to average the measured atmospheric $CH_4$ and $\delta^{13}CH_4$ values to 5 or 10 minutes instead of 1min and found no decrease in the error of the determined mean isotopic source signatures. |

| [Referee#1] | Line 271: First you said it is not possible to get reliable results on diurnal cycle then in line 271 you say, "This indicates that the composition of $CH_4$ sources in Heidelberg is the same during day and night". It seems to be contradictory. Please clarify, also regarding impact of the instrument precision for diurnal measurements. |
|---|---|
| [Hoheisel and Schmidt] | The mean isotopic source signature determined with the moving Miller-Tans/Keeling plot approach does not provide us with results to reliably resolve diurnal cycles. A higher precision of the instrument would make it possible to obtain mean isotopic signatures even for small $CH_4$ ranges with small fitting errors. Thus, more mean isotopic source signatures will match the chosen selection criteria for an interval of a few hours, and thus a higher temporal resolution is possible. |
| | Later in the manuscript, we compared the monthly mean isotopic source signatures obtained with the night-time and the moving Miller-Tans/Keeling plot approach. The first approach uses only the nighttime measurements, while the second approach uses daytime and nighttime data. As we could not find a significant difference between the two methods, we concluded that this could be caused by two possibilities: First, there is no difference in the composition of emissions between day and night, and second, the moving Miller-Tans/Keeling plot approach is influenced mostly by the nighttime increase. |

| [Referee#1] | Line 288: Do the monthly approach and moving Miller-Tans approach represent the same catchment area (both bigger than night-time approach? If yes, this hypothesis does not explain differences in results from monthly and moving Miller Tans approaches. Please comment. |
|---|---|
| [Hoheisel and Schmidt] | It is likely that the monthly approach and the moving Miller-Tans/Keeling plot approach do not represent the same catchment area, which may explain the differences observed in the two approaches. |

| [Referee#1] | In table 1., $\delta^{13}C$-$CH_4$ for road transport comes from Levin et al. 1993. Is it possible this value changed over last 30 years as different cars are used now and then (e.g. diesel versus petrol, better technology etc)? Is it possible the inventories results are biased comparing to atmospheric results due to unaware shift between used $\delta13C$-$CH4$ from previous studies and real values? |
|---|---|
| [Hoheisel and Schmidt] | Yes, it is true that the composition of diesel compared to gasoline and the technology in cars has changed over the last 30 years. Therefore, the $\delta^{13}C$-$CH_4$ value for road transportation may also have changed since then. Unfortunately, there are only a few new studies on $\delta^{13}C$-$CH_4$ measurements on car exhaust gases in Europe. The few values range between -20‰ (Menoud et al., 2022 measured in Hamburg in 2022) and -28‰ (Levin et al., determined in Heidelberg in 1999). We assume that the $13CH4$ values of car exhaust gases are in this range. And thus, we chose the value of -22.8‰, which was reported by Levin et al. (1993). Chanton et al. (2000) describe $\delta^{13}C$-$CH_4$ values between -22‰ and -9‰ for 16 vehicles sampled in the United State. The value of -22.8‰ chosen by us is in the lower range. |
| | If we assume that the $\delta^{13}C$-$CH_4$ value for road transportation is -9‰ or -28‰ instead of -22.8‰, the annual mean isotopic carbon signatures, determined from LUBW/EDGAR data, change by 0.38‰/0.08‰ or 0.04‰/0.03‰. This is due to the fact that $CH_4$ emissions from road traffic are only a small part of the total $CH_4$ emissions of 2.1% and 0.6% in the LUBW and EDGAR inventory. Although the actual $\delta^{13}C$-$CH_4$ value for traffic today may differ from the value used in this study, this has no essential impact on our results. |

| [Referee#1] | Line 295-297: "the nearby $CH_4$ sources are more often natural gas leaks, wastewater, traffic, or emissions from energy for buildings. These $CH_4$ emissions are on average less depleted." – it sounds like wastewater is also less depleted, in the same category as other mentioned sources. Based on Tab 1, it is clear they are more depleted, as other microbial sources. |
|---|---|
| [Hoheisel and Schmidt] | Yes, this sentence was misleading. We changed it in the revised manuscript. |

| [Referee#1] | Line 390: What is the difference between value from Widory et al. (2006) and "publications describing $^{13}C$ for $CH_4$ emitted by waste incineration in the way we needed them to calculate the mean $\delta^{13}C$-$CH_4$ isotopic source signature" What is the "needed way" and how it is different from method presented in Widory et al. (2006)? |
|---|---|
| [Hoheisel and Schmidt] | During the development process of the paper, the paragraph in line 390 changed several times, so that in the end it unfortunately became misleading and imprecise. We just wanted to express that despite extensive literature research we have not found a study that has determined and reported the $\delta^{13}C$-$CH_4$ isotope signature of waste incineration. Thus, we adopted the $^{13}C$ composition of waste incineration reported by Widory et al. (2006) for $CO_2$. We have changed the sentence in the revised manuscript. |

| [Referee#1] | Line 415-420: Repeating annual mean results here brings some confusion. I suggest removing it and focus only on annual cycle in this paragraph. |
|---|---|
| [Hoheisel and Schmidt] | Thank you for your suggestions. We have slightly changed the structure of the subchapter in the revised manuscript to make it easier to follow. |

**Technical corrections:**

| [Referee#1] | Line 30 and further: δ13C-CH4 should be given in order from smaller to bigger, e.g., (-70 ‰ to -55 ‰) instead of (-55 ‰ to -70 ‰). |
|---|---|
| [Hoheisel and Schmidt] | Thanks, we changed it in the revised manuscript. |

| [Referee#1] | Figure 9: Remove too at the end of last sentence |
|---|---|
| [Hoheisel and Schmidt] | Yes, we removed it in the revised manuscript. |

| [Referee#1] | The link to access used data does not work. |
|---|---|
| [Hoheisel and Schmidt] | We are still working on the public permanent DOI to the measured Heidelberg data. So far, we have provided a preliminary DOI for the editor and the reviewers to the editor. |

---

## Author Comment (AC2)

**Author's Response to Referee#2 Comments on:** "Six years of continuous carbon isotope composition measurements of methane in Heidelberg (Germany) – a study of source contributions and comparison to emission inventories"

We thank the Referee#2 for the careful reading and appreciate the referee's suggestions. These helped us improve the manuscript.

**General Comments:**

| | |
|---|---|
| [Referee#2] | In this paper, Hoheisel et Schmidt describe new continuous CH4 and δ(13C,CH4) measurements retrieved between 2014 and 2020 In Heidelberg (Germany). After introducing the experimental setup, they analyze the temporal variability of this data and apply the Miller-Tans method to derive estimates of the mean isotopic signature that could cause these variations. These determined estimates are then compared to bottom-up estimates using two different inventories. |
| | Overall, the paper is well presented and well written. The structure is clear and it is easy to understand where the authors are leading us. Also, the scientific questions addressed in this study are well within the scope of ACP and the analysis conducted to answer these questions is detailed, elaborate and tackles very interesting points, both for experimentalists and atmospheric modelers. |
| | Last but not least, this new continuous data is invaluable to better investigate methane sources and will likely be utilized in the future by the rest of the atmospheric community. Most of my comments only call for additional clarity in the methodology and the presentation of results. Also, a few additional details in the methodology and in the results would be beneficial both for the reproducibility of the study and the comprehensiveness of the analysis. However, these comments are very minor and I can already recommend this paper for a publication in the journal ACP. |

**Specific comments:**

| | |
|---|---|
| [Referee#2] | Line 1: I recommend not using the abbreviation $\delta(^{13}CH_4)$ in the abstract. Use $\delta(^{13}C,CH_4)$. |
| [Hoheisel and Schmidt] | Yes, we changed $\delta(^{13}CH_4)$ to $\delta(^{13}C,CH_4)$ in the abstract. |

| | |
|---|---|
| [Referee#2] | Line 2: You write that it is a 6-year time-series, since 2014. It may give the reader the impression that the measurements stopped in 2020. I suggest a small revision: "Between 2014 and 2020, the time series shows an increasing trend of $(6.8 \pm 0.3)$ nmolmol$^{-1}$ a$^{-1}$ for the CH4 mole fraction." |
| [Hoheisel and Schmidt] | Thank you for the helpful suggestion. This will make the text more comprehensible. We changed the sentence in the revised manuscript. |

| | |
|---|---|
| [Referee#2] | Line 6: At present, it seems you are using $\delta(^{13}C,CH_4)$ (abbreviated as $\delta(^{13}CH_4)$) for atmospheric isotopic composition and $\delta^{13}C$ (also an abbreviation of $\delta(^{13}C,CH_4)$) for isotopic signature. In my opinion, it's okay to keep it that way but you should not use the abbreviations in the abstract and also introduce the abbreviation $\delta^{13}C$ in the main text. |
| | Line 166: You should very briefly introduce the abbreviation δ13C the same way you introduced it for $\delta(^{13}CH_4)$ for the atmospheric isotopic composition. |
| [Hoheisel and Schmidt] | Thank you for pointing this out. To avoid confusion, we have replaced the term '$\delta^{13}C$ isotopic source signature' with 'isotopic carbon source signature'. |

| | |
|---|---|
|  | Line 9: Sentence not clear. As far as I understand, the mean estimated δ($^{13}$C,CH$_4$) source isotopic signature exhibits a seasonal variation, with a peak-to-peak variation -6.2 ‰. If it is the case, you should reformulate. I suggest replacing the sentence "This annual cycle in $^{13}$C-CH$_4$ sources…", with "This annual cycle in the mean source isotopic signature source, with a peak-to-peak amplitude of −6.2‰, can only be partially explained by seasonal variations in the $^{13}$C-enriched emissions from heating." |
| [Hoheisel and Schmidt] | Yes, you understand it right. We changed the sentence in the revised manuscript according to your suggestion. |

| | |
|---|---|
| [Referee#2] | Line 25: After this sentence, you should introduce the isotopic scale δ($^{13}$C,CH$_4$) with a formula. Because after that you provide typical source signature values for different source categories (e.g. −55‰ to −70‰), but the reader does not know what scale you are referring to. You could also be talking about $^{14}$C rather than $^{13}$C. |
| | Line 99: Would be worth mentioning the value you are using for the reference $^{13}$C/$^{12}$C ratio because there is sometimes confusion between PDB and VPDB values. |
| [Hoheisel and Schmidt] | As suggested, we included an introduction to the δ –notation and the VPDB values in the revised manuscript: |
| | *"The isotopic composition of methane δ($^{13}$C,CH$_4$), hereafter abbreviated as δ($^{13}$CH$_4$), is described with the δ –notation, using the isotopic ratio R, and is typically given in ‰. The international reference standard for δ($^{13}$CH$_4$) is the Vienna Pee Dee Belemnite (VPDB; 0.0111802±0.0000028, Werner and Brand, 2001).* |
| | $$\delta = R_{sample}/R_{standard} - 1; \quad R = {}^{13}CH_4/{}^{12}CH_4"$$ |

| | |
|---|---|
| [Referee#2] | Line 30: Give the recent references for these values also here (e.g., Sherwood et al. 2017; 2021; Menoud et al., 2022). Also, the thermogenic range you provide appears slightly inconsistent (too small and too enriched) when compared to the information presented in these references. |
| [Hoheisel and Schmidt] | This paragraph in the introduction is intended to give an overview of the different CH$_4$ sources and their typical isotopic source signatures. As indicated in the manuscript, we have taken the typical isotopic source signatures from the IPCC AR5 WG1 report from 2013. Individual measurements and especially more recent measurements may of course deviate from these typical values. To avoid any misunderstandings, we have changed the paragraph and added a sentence that discusses the more recent measurement results. |
| | "*CH$_4$ is emitted from anthropogenic and natural sources, which are grouped in three different categories according to the production processes. Biogenic CH$_4$ is produced under anaerobic conditions due to degradation of organic matter (typically -70‰ to -55‰; IPCC, 2013). Biogenic CH$_4$ sources are wetlands, ruminants, landfills and wastewater treatment plants. Thermogenic CH$_4$, like that in natural gas, is formed on geological time scales out of organic matter and is less depleted than biogenic CH$_4$ (typically -45‰ to -25‰; IPCC, 2013). Pyrogenic CH$_4$ is formed during the incomplete combustion of organic matter, such as biomass burning, and is more enriched (typically -25‰ to -13‰; IPCC, 2013) compared to biogenic and thermogenic CH$_4$. Studies by Sherwood et al. (2017; 2021) and Menoud et al. (2022) show that the δ$^{13}$C-CH$_4$ values of the different source categories are not always as distinct as indicated above. They give much larger ranges of δ$^{13}$C-CH$_4$ values for the different source categories, which also overlap as a result. Especially for fossil but also for biogenic sources large regional differences occur.*" |

| | |
|---|---|
| [Referee#2] | Line 59: "To our knowledge, our time series is the longest in situ $\delta(^{13}CH_4)$ record, with high temporal resolution, reported to date.". You mean the longest for Heidelberg or the longest ever in the world? |
| [Hoheisel and Schmidt] | We mean, that to our knowledge our time series is the longest published time series measured with a high temporal resolution instruments in the world. |

| | |
|---|---|
| [Referee#2] | Line 64: A continuous six-year time series between when and when? |
| [Hoheisel and Schmidt] | We included the measurement years in the revised manuscript. |

| | |
|---|---|
| [Referee#2] | Line 119: I think two significant figures is not enough for $\delta(^{13}CH_4)$. I would recommend three, as you do in the rest of the paper. If we look at the bulk, I mostly see values between -49.0‰ and -47.5‰. If you also want to include extreme values, then it is approximately -49.5‰ to -47.2‰ |
| [Hoheisel and Schmidt] | As recommended, we changed the number of significant figures to three for $\delta(^{13}CH_4)$ in the revised manuscript:
 "*The corresponding isotopic composition $\delta(^{13}CH_4)$ ranges from -49.3‰ to -47.3‰.*" |

| | |
|---|---|
| [Referee#2] | Line 123: Please plot the trend on Figure 3. I think if you increase the size of the figure and increase the transparency of the 1day averages, then it won't decrease the overall clarity. |
| [Hoheisel and Schmidt] | As suggested, we have added the trend in Figure 3 of the revised manuscript. |

| | |
|---|---|
| [Referee#2] | Line 127: Less enriched compared to what? To the mean? Overall, I think some confusion can arise from the fact that you use the mean over the full time series as the center of your annual cycle, rather than using zero. Line 128 and 129 suggest that the values are always -48.3‰ in early autumn and -47.9‰ in spring, while these values have an interannual variability, due to the trend and the variations in the seasonal cycle. It's okay to keep it that way but you should comment on that. |
| [Hoheisel and Schmidt] | Thank you for your comment. We have chosen to add the mean values when presenting the mean annual cycles, as in our opinion this enables a more intuitive understanding of the annual cycles. However, we agree with you that stating the maximum and minimum values in the text, in the way we have done it, can lead to confusion and misunderstandings. We have therefore changed the sentence in the revised manuscript:
 "*The annual cycle in atmospheric $\delta(^{13}CH_4)$ has a mean amplitude of 0.4‰. In early autumn (September to October) the $\delta(^{13}CH_4)$ values are more depleted than the values in spring (April to May).*" |

| | |
|---|---|
| [Referee#2] | Line 139: After the analysis in Section 3.3, can you think of a reasonable explanation? |
| [Hoheisel and Schmidt] | The stronger influence of biogenic emissions in summer could lead to patterns in the measured $\delta(^{13}CH_4)$ values in Heidelberg, such as the lower values in fall or the greater amplitude in the diurnal cycle in summer. However, other factors such as the OH sink also play a role in the measured atmospheric $\delta(^{13}CH_4)$ values, so we do not want to over interpret the influence on the measured $\delta(^{13}CH_4)$ values. |

| [Referee#2] | Line 146: Could you provide the details of the Mace Head Observatory (altitude, longitude, latitude)? Also, please explain why Mace Head can be considered as a "background" station. |
|---|---|
| [Hoheisel and Schmidt] | We included more details about the Mace Head Observatory in the revised manuscript: "*The Mace Head Observatory (53°19'36"N, 9°54'16"E, 8.4m a.s.l.) is located on the west-coast of Ireland and measures the maritime background mole fraction when air is coming from the ocean.*"

We are aware that Mace Head is not optimal as a background station for Heidelberg. However, since this is the only background station west of Heidelberg with a long published $\delta^{13}$C-CH$_4$ record, we have decided to use the data from Mace Head to characterise and compare the Heidelberg measurements of today (2014-2020) and the 1990s. |
| [Referee#2] | Line 156: In this paragraph, you use both "emissions in Heidelberg" and "emissions in the catchment area of Heidelberg". Is it supposed to mean the same thing (I suppose so)? Or did Levin et al. (2011, 2021) only analyze emissions in the city of Heidelberg (without the surroundings)? |
| [Hoheisel and Schmidt] | Yes, we meant the same with the terms "emissions in Heidelberg" and "emissions in the catchment area of Heidelberg". Thank you for pointing out this inaccuracy. We corrected this in the revised manuscript. |
| [Referee#2] | Line 182: If you want to make a comparison between the Miller-Tans method and the Keeling plot, which is a good idea, you should briefly introduce the Keeling plot method as well. |
| [Hoheisel and Schmidt] | Thank you for pointing this out. We have changed large parts of subsection 3.3.1 on the Miller-Tans and Keeling plot methods in the revised manuscript to clarify the methods we use based on Referee#1's comments. We also included a short introduction to the Keeling plot and the Miller-Tans methods. |
| [Referee#2] | Line 187 and line 200: Which CH$_4$ range? Apologies for being confused here. You discard every data point where the difference between C$_{bg}$ and C$_{obs}$ is below 100 nmol mol-1? If it is the case, for the night data set, do you discard the full night if one of the data points does not satisfy this criteria? |
| [Hoheisel and Schmidt] | Thank you for this helpful comment. We have changed the paragraph in the revised manuscript to clarify our selection criteria. With 'CH$_4$ range', we did not mean the difference between C$_{bg}$ and C$_{obs}$, but the difference between the minimum and maximum observed CH$_4$ values (C$_{obs}$).

"*The uncertainty of the source signature determined with the Keeling plot method and the York fit strongly depends on the precision of the analyser and the peak height of CH$_4$ (Hoheisel et al., 2019). To achieve accurate results for the mean isotopic carbon source signatures, we apply two criteria to our data: the CH$_4$ range of the dataset, to which the Keeling plot is applied, has to be larger than 100 nmol mol$^{-1}$ and the fit error on the slope of the regression line has to be smaller than 2.5‰.*" |
| [Referee#2] | Line 193: Please, reformulate. What can be assumed? That it is constant? |
| [Hoheisel and Schmidt] | We reformulated the sentence in the revised manuscript:

"*However, for such large time intervals of one month, the assumption of a constant background, which is used in the Keeling plot method, is not correct and could lead to errors in the determined isotopic source signatures. This problem is less prominent for the night-time and moving Keeling plot approach, as in these approaches shorter time periods of a few hours are used.*" |

| [Referee#2] | Line 213: What do you mean by 'directly adjacent in time'? What amount of time do you consider to be 'adjacent in time'? And what percentage of the 18% are during night time?
In general, I do not understand how long an "event" is. Therefore, it is difficult to confirm that a night time increase can influence the event detected by the Miller-Tans approach. It would be nice to show somewhere the typical length of an event. Do these events happen mostly during the night? As far as I understand, you can access this information with your methodology. |
|---|---|
| [Hoheisel and Schmidt] | 53% of the 18% mean isotopic source signatures of the moving Miller-Tans/Keeling plot approach which met our criteria are during the night.
Based on your comments and the ones from Referee#1, we had a closer look at the automatically generated events and decided not to split the moving Miller-Tans/Keeling plot results into individual events any more. Instead, we will examine the hourly and daily averages of the results of the moving Keeling plot. We changed this in the revised manuscript accordingly. |

| [Referee#2] | Line 222: Here and throughout the text, you often use "more/less depleted" or "more/less enriched". Usually, a value is depleted/enriched compared to a point of comparison, which is often the atmospheric value (around -48.07‰ in your case), as you state it very clearly in the following sentence. For instance, in this situation, I would use "more depleted" rather than "less enriched" because the source signatures for the moving Miller-Tans and night time approaches are already depleted compared to the atmospheric composition. Therefore, the third one is even more depleted.
Line 292 and onwards: Again "more depleted" compared to what? |
|---|---|
| [Hoheisel and Schmidt] | Thank you for this important comment. Unfortunately, we missed this inaccuracy while writing the manuscript. We went through the entire document again and tried to formulate the comparison of isotopic source signatures and $\delta^{13}C\text{-}CH_4$ more precisely. |

| [Referee#2] | Line 224: It seems that you are suggesting there is a causality between the fact that the estimated source isotopic signature is more depleted than the atmospheric composition and the fact that biogenic sources play a dominant role. I do not think it is true. You can suggest biogenic sources are dominant because the estimated source signature is low and close to what could be expected if biogenic sources (typically between -55‰ and -70‰) were dominant. Or to the contrary, it would be too low compared to a situation where only pyrogenic and thermogenic were dominant (although some thermogenic sources can have a source isotopic signature as low as -60‰, see Sherwood et al., 2017). |
|---|---|
| [Hoheisel and Schmidt] | Thank you again for pointing out inaccuracies and imprecision in our text. We corrected the sentence according to your suggestions in the revised manuscript:
*"Since the determined mean isotopic source signature is low and close to what could be expected if biogenic sources (typically between -55‰ and -70‰) were dominant, a strong influence from biogenic $CH_4$ sources, such as waste management and agriculture, in the catchment area of Heidelberg can be assumed."* |

| [Referee#2] | Line 271: Please reformulate. Do you mean both explanations are plausible? Is it exclusive? The too at the end is a bit misleading. |
|---|---|
| [Hoheisel and Schmidt] | Yes, both explanations are plausible. We changed the sentence in the revised manuscript. |

| | |
|---|---|
| [Referee#2] | Line 280: You suggest two explanations but as far as I understand, both explanations are closely linked. Small pollution events of the first explanation can be the ones from distant sources from the second explanation. If it is correct, maybe you could mention it at the end of the paragraph. |
| [Hoheisel and Schmidt] | Thank you for this constructive comment. We agree that the paragraph is not clear enough and have reworked it slightly in the revised manuscript. |

| | |
|---|---|
| [Referee#2] | Line 303: "The monthly values vary on average between 0.1‰ and 0.8‰". What does the percentage represent? |
| [Hoheisel and Schmidt] | Thank you for your comment. We noticed, that this sentence does not clearly describe the deviations which occur, when using different selection criteria. The values 0.1‰ and 0.8‰ represent differences in the monthly mean isotopic source signatures calculated using different selection criteria. We changed the sentence in the revised manucript: *"The monthly mean isotopic source signatures calculated with different selection criteria show differences between 0.1‰ and 0.8‰, with standard deviations between 1‰ and 3‰."* |

| | |
|---|---|
| [Referee#2] | Line 319: Give the exact location of the station. |
| [Hoheisel and Schmidt] | We included the location of the Schauinsland station in the revised manuscript: 47°54'50" N, 7°54'28" E, 1205m a.s.l. |

| | |
|---|---|
| [Referee#2] | Line 340: You have only one subsection 3.4.1 under section 3.4. Shouldn't section 3.5 and 3.6 be subsections of sections 3.4? Or at least section 3.5? |
| [Hoheisel and Schmidt] | Yes you are right, thank you very much. Section 3.5 and 3.6 should be subsections of 3.4. We changed it in the revised manuscript. |

| | |
|---|---|
| [Referee#2] | Line 354: Why does it "seem" to decrease? If you are not confident, where does this number 7%, without any uncertainty, come from? |
| [Hoheisel and Schmidt] | Thanks. We have corrected the sentence in the revised manuscript. |

| | |
|---|---|
| [Referee#2] | Line 389: It is not clear what you needed. |
| [Hoheisel and Schmidt] | During the development process of the paper, the paragraph in line 389 changed several times, so that in the end it unfortunately became misleading and imprecise. We only wanted to express that, despite extensive literature research, we have not found a study that has determined and reported the $\delta^{13}C$-$CH_4$ isotope signature of waste incineration. We have changed the sentence in the revised manuscript. |

**Technical comments:**

| | |
|---|---|
| [Referee#2] | Figures: Although they have a good resolution, I would have preferred the figures to be larger, i.e. fitting the width of the page. |
| [Hoheisel and Schmidt] | This is a good point. The reason we chose this width is to comply with the guidelines of the journal. Hence, if the editor agrees I would be happy to enlarge the width. |

[Referee#2]  Figure 1: In the caption, map data on from → map data from

[Hoheisel and Schmidt]  We corrected it in the revised manuscript.

[Referee#2]  Figure 3: There is a problem with the x-axis ticks of the top-right panel. Please, make it similar to the bottom-right panel. Also, please add in the caption a note saying that the y-axis ranges are not the same for the left and right panels.

[Hoheisel and Schmidt]  Many thanks for the hint. We have corrected the figure and added a sentence regarding the y-axis ranges in the revised manuscript.

[Referee#2]  Line 67: to $CH_4$ total emissions.

[Hoheisel and Schmidt]  We corrected it in the revised manuscript.

[Referee#2]  Line 152: Replace with "This was different" or "this is different for the 1990s"

[Hoheisel and Schmidt]  We corrected it in the revised manuscript.

[Referee#2]  Line 180: Put these subscripts in the same format as in the equation.

[Hoheisel and Schmidt]  We changed it in the revised manuscript.

[Referee#2]  Line 214: Do you mean 2014 instead of 2011?

[Hoheisel and Schmidt]  Yes exactly, we meant 2014. We corrected it in the revised manuscript.

[Referee#2]  Line 266: Although → However

[Hoheisel and Schmidt]  We changed it in the revised manuscript.

[Referee#2]  Line 274: less enriched → more depleted

[Hoheisel and Schmidt]  We changed it in the revised manuscript.

[Referee#2]  Line 294: remove the space between the first parenthesis and "see"

[Hoheisel and Schmidt]  We corrected it in the revised manuscript.

[Referee#2]  Line 331: go one step further → extend the effort?

[Hoheisel and Schmidt]  We changed it in the revised manuscript.

[Referee#2]  Line 378: This is supported by the fact that the amount of emissions from sectors, such as livestock farming, with well studied emission factors and accurate statistical data are comparable for both inventories → This is supported by the fact that the amount of emissions from sectors with well studied emission factors and accurate statistical data are comparable for both inventories, such as livestock farming, are comparable for both inventories.

[Hoheisel and Schmidt]  We corrected this in the revised manuscript.

---

## Author Response (AR2)

**Author's Response to Editor Comments on:** "Six years of continuous carbon isotope composition measurements of methane in Heidelberg (Germany) – a study of source contributions and comparison to emission inventories"

We thank the editor for the detailed and constructive comments. We will revise the manuscript according.

**General Comments:**

[Editor] Dear Dr Hoheisel
Many thanks for your revised manuscript.
I'd be happy to accept it for publication in Atmospheric Chemistry and Physics subject to the revisions detailed below, which I will oversee myself.
Sincerely
Jan Kaiser Editor ACP

[Editor] Abstract: The abstract should be revised in line with the guidelines detailed here:https://www.atmospheric-chemistry-and-physics.net/policies/guidelines_for_authors.html
Please ensure you identify the status of scientific understanding, the gap in knowledge being addressed and comment on the importance and implications of the results.

[Hoheisel and Schmidt] We revised the abstract in accordance with the above guidance.

[Editor] Conclusion: The conclusions do not make clear the implications of the results mean for our understanding of the state and/or behaviour of the atmosphere and climate. These implications and the revised abstract will help me decide whether the manuscript can be designated a "Research article" or whether it should be a "Measurement report" (https://www.atmospheric-chemistry-and-physics.net/about/manuscript_types.html).

[Hoheisel and Schmidt] We have revised the conclusion to better highlight the implications of this study. Emphasis has been placed both on pointing out differences with the EDGAR emission inventory and on the annual variation of biogenic $CH_4$ emissions. In particular, this annual cycle of $CH_4$ emissions has not been shown in other atmospheric studies and can therefore make a unique contribution to the understanding of the $CH_4$ cycle.

**Technical corrections and minor revisions:**

[Editor] Eq 1: Please write the letter R in italics and define R using appropriate quantity symbols, e.g. R = n(13CH4)/n(12CH4) or R = C(13CH4)/C(12CH4).

[Hoheisel and Schmidt] We have changed Eq. 1 in the revised manuscript.

[Editor] l. 85, 156 and 338: Please add a space before "N", "E" and "m" and spell out "above sea level".

[Hoheisel and Schmidt] We included the suggestions in the revised manuscript.

[Editor] Figure widths: Please make all figures double-column width (16 cm) except Fig. 9. The copy-editing team should be able to help with this.

[Hoheisel and Schmidt] We changed the figure widths in the revised manuscript as suggested by the editor.

[Editor] Data availability: Please ensure all data are in the online repository and the DOI is working before uploading the revised manuscript. The data must be available permanently and not on request. Therefore, the text "and on request from the data owner (martina.schmidt@iup.uni-heidelberg.de)" should not be necessary and be removed.

[Hoheisel and Schmidt] The data will be accessible from 19 January 2024 with the DOI (https://doi.org/10.11588/data/OXKVW2) on heiDATA, the institutional repository for Open Research Data from Heidelberg University.

---

## Author Response (AR3)

**Author's Response to Editor Comments on:** "Six years of continuous carbon isotope composition measurements of methane in Heidelberg (Germany) – a study of source contributions and comparison to emission inventories"

We thank the editor for accepting the manuscript for publication in Atmospheric Chemistry and Physics, subject to a few technical corrections. We will incorporate these corrections in the revised manuscript.

**General Comments:**

[Editor]
Dear Dr Hoheisel
Many thanks for your revised manuscript, which I am happy to accept it for publication in Atmospheric Chemistry and Physics subject to a couple of technical corrections:

l. 30/Eq. 1: Chemical symbols cannot be subject to mathematical operations, so please use an appropriate physical quantity symbol to write the isotope ratio, e.g. $R = N(13C)/N(12C)$ [with N in italics; representing the quantity "number"], $R = n(13C)/n(12C)$ [with n in italics; representing the quantity "(chemical) amount", or $R = C(13C)/C(12C)$ [with C in italics; with C representing amount fractions/mole fractions].

l. 464: the -> a, i.e. "a continuous time series"

Thank you for your contribution to Atmospheric Chemistry and Physics.

Sincerely
Jan Kaiser
Editor ACP

**Technical corrections:**

[Editor]                                 l. 30/Eq. 1
[Hoheisel and Schmidt]    Please excuse that we have not stated the equation formally correctly. To correct this, we have slightly changed the paragraph in the revised manuscript.

[Editor]                                 l. 464
[Hoheisel and Schmidt]    We have corrected it in the revised manuscript.

[Hoheisel and Schmidt]    In the acknowledgements we have included the sentence:
*For the publication fee we acknowledge financial support by Deutsche Forschungsgemeinschaft within the funding programme „Open Access Publikationskosten" as well as by Heidelberg University.*